# Elastic instability during branchial ectoderm development causes folding of the *Chlamydosaurus* erectile frill

Sophie A Montandon[1†], Anamarija Fofonjka[1,2†], Michel C Milinkovitch[1,2]*

[1]Laboratory of Artificial & Natural Evolution (LANE), Department of Genetics & Evolution, University of Geneva, Geneva, Switzerland; [2]SIB Swiss Institute of Bioinformatics, Geneva, Switzerland

**Abstract** We study the morphogenesis and evolutionary origin of the spectacular erectile ruff of the frilled dragon (*Chlamydosaurus kingii*). Our comparative developmental analyses of multiple species suggest that the ancestor of Episquamata reptiles developed a neck fold from the hyoid branchial arch by preventing it to fully fuse with posterior arches. We also show that the *Chlamydosaurus* embryonic neck fold dramatically enlarges and its anterior surface wrinkles, establishing three convex ridges on each lobe of the frill. We suggest that this robust folding pattern is not due to localised increased growth at the positions of the ridges, but emerges from an elastic instability during homogeneous growth of the frill skin frustrated by its attachment to adjacent tissues. Our physical analog experiments and 3D computational simulations, using realistic embryonic tissue growth, thickness and stiffness values, recapitulate the transition from two to three ridges observed during embryonic development of the dragon's frill.
DOI: https://doi.org/10.7554/eLife.44455.001

*For correspondence:
michel.milinkovitch@unige.ch

†These authors contributed equally to this work

Competing interests: The authors declare that no competing interests exist.

## Introduction

Lizards can exhibit moveable skin folds at various locations of their body, such as the wings of the flying dragon (*Draco volans*), the oral display frill of the 'secret toadhead agama' (*Phrynocephalus mystaceus*), and the dewlap of many anole lizard species (*Anolis* spp.). Here, we investigate the evolutionary developmental origin of the distinctive large erectile ruff (*Figure 1A,B*) of the emblematic Australian/New-Guinean frilled dragon (*Chlamydosaurus kingii*). This animal spreads its spectacular neck frill for predator deterrence, territorial display and courtship (*Shine, 1990*). *Figure 1C* illustrates that the ventral sides of the ruff are supported by the two ceratobranchial I bones (CBI) of the hyoid apparatus (*Beddard, 1905*) and the dorsal sides are held by the so-called 'Grey's cartilages' (*De Vis, 1883*). Erection of the frill is caused by the coordinated movements of the CBI bones and Grey's cartilages and requires the opening of the mouth.

Although their primary function in tetrapods is associated with deglutition, the bones of the hyoid apparatus are also involved in a variety of specialised morphologies and functions such as improved lung ventilation through gular pumping in monitor lizards (*Bels et al., 1995*; *Owerkowicz et al., 1999*), extension of the throat in bearded dragons (*Throckmorton et al., 1985*) and of the dewlap in *Anolis* lizards (*Bels, 1990*; *Font and Rome, 1990*), tongue projection in chameleons (*Herrel et al., 2001*), as well as tongue extension and shock absorption in woodpeckers (*Yoon and Park, 2011*). In reptiles, the central part of the hyoid apparatus, anteriorly prolonged by an entoglossal process (EP; *Figure 1C–D*), is associated to three pairs of horns (*Bellairs and Kamal, 1981*). The first pair is composed of the hypohyal (HH), ceratohyal (CH) and epihyal (EH). The second pair is made of the CBI and epibranchial (EB), while the third pair of horns consists of the ceratobranchial II (CBII). During embryogenesis, the hyoid apparatus develops from the pre-cartilage (mesenchyme

**eLife digest** In Jurassic Park, while the computer programmer Dennis Nedry attempts to smuggle dinosaur embryos off the island, he gets attacked and killed by a mid-sized dinosaur that erects a frightening neck frill. This fictional dinosaur is clearly inspired from a real animal known as the 'frilled dragon', that lives today in northern Australia and southern New Guinea.

These lizards, also known as *Chlamydosaurus kingii,* have a large disc of skin that sits around their head and neck. This frill is usually folded back against the body, but can spread in a spectacular fashion to scare off predators and competitors. Folding of the left and right side of the frill occurs at three pre-formed ridges. But, it remains unclear which ancestral structure evolved to become the dragon's frill, and how the ridges in the frill form during development.

Now, Montandon, Fofonjka, and Milinkovitch show that the dragon's frill, as well as the bone and cartilage that support it, develop from a part of the embryo known as the branchial arches. These are a series of bands of tissue in the embryo that evolved to become the gill supports in fish, and that now give rise to multiple structures in the ear and neck of land vertebrates. In most species, the second branchial arch will eventually fuse with the arches behind it. But in the frilled dragon, this arch instead continues to expand, leading to the formation of the dragon's spectacular frill.

As the frill develops, the front side of the skin forms three successive folds, which make up the pre-formed ridges. Studying the formation of these ridges revealed that they do not emerge from increased growth at the folding sites, but from physical forces – whereby the growth of the frill is constrained by its attachment to the neck. This causes the top layer to buckle, creating the folds of the frill. Montandon, Fofonjka, and Milinkovitch then simulated this mechanism of growth in a computer model and found it could recapitulate how folds develop in the frill of real lizard embryos.

These results provide further evidence that physical processes, as well as genetic programs, can shape tissues and organs during an embryo's development. Furthermore, changes in how the branchial arches develop between lizard species highlights how evolution is able to 'recycle' old structures into new shapes with different roles.

DOI: https://doi.org/10.7554/eLife.44455.002

condensation of neural crest origin) of three branchial arches (BA): the hyoid arch (*i.e.*, the second BA = BA2) contributes to the development of the central and anterior parts of the hyoid body as well as the first pair of horns, whereas the third and fourth BAs (BA3 and BA4) generate the second and third pairs of horns, respectively (*Bellairs and Kamal, 1981*; *Creuzet et al., 2005*; *Kaufman and Bard, 1999*; *Köntges and Lumsden, 1996*).

Contrary to that of the hyoid skeletal elements, the morphogenesis of the frill soft tissues and of the 'Grey's cartilage' are unknown. Here, using computed-tomography and histology approaches, we first show that the highly-developed CBI bones of the frilled dragon are localised into the third (most dorsal) skin ridge of the frill and that the 'Grey's cartilage' is not made of cartilage per se, but is a dense connective tissue mainly composed of collagen fibres. Second, our comparative developmental analyses indicate that the existence of a spectacular frill in *Chlamydosaurus* was made possible by the incomplete fusion of the BA2 with the cardiac eminence and posterior BAs, an evolutionary event that probably occurred at the origin of Episquamata reptiles. This event allowed most members of that lineage to exhibit a conspicuous neck fold (although it was lost in chameleons, snakes and legless lizards) that develops from the hyoid BA (BA2). Hence, the *Chlamydosaurus* frill is a dramatic outgrowth of the hyoid arch ectoderm.

Finally, using 3D reconstruction, analyses of proliferation and computational simulations, we show that the very robust folding pattern of the *Chlamydosaurus* frill (all individuals develop three ridges on each of the two lobes of the frill) is not due to localised increased growth at the position of the ridges, but likely emerges from an elastic instability during the homogeneous growth of the anterior sheet frustrated by the underlying tissues and by its attachment to the neck. This physical (mechanical) process also explains the transition from two to three ridges observed during embryonic development of the dragon frill.

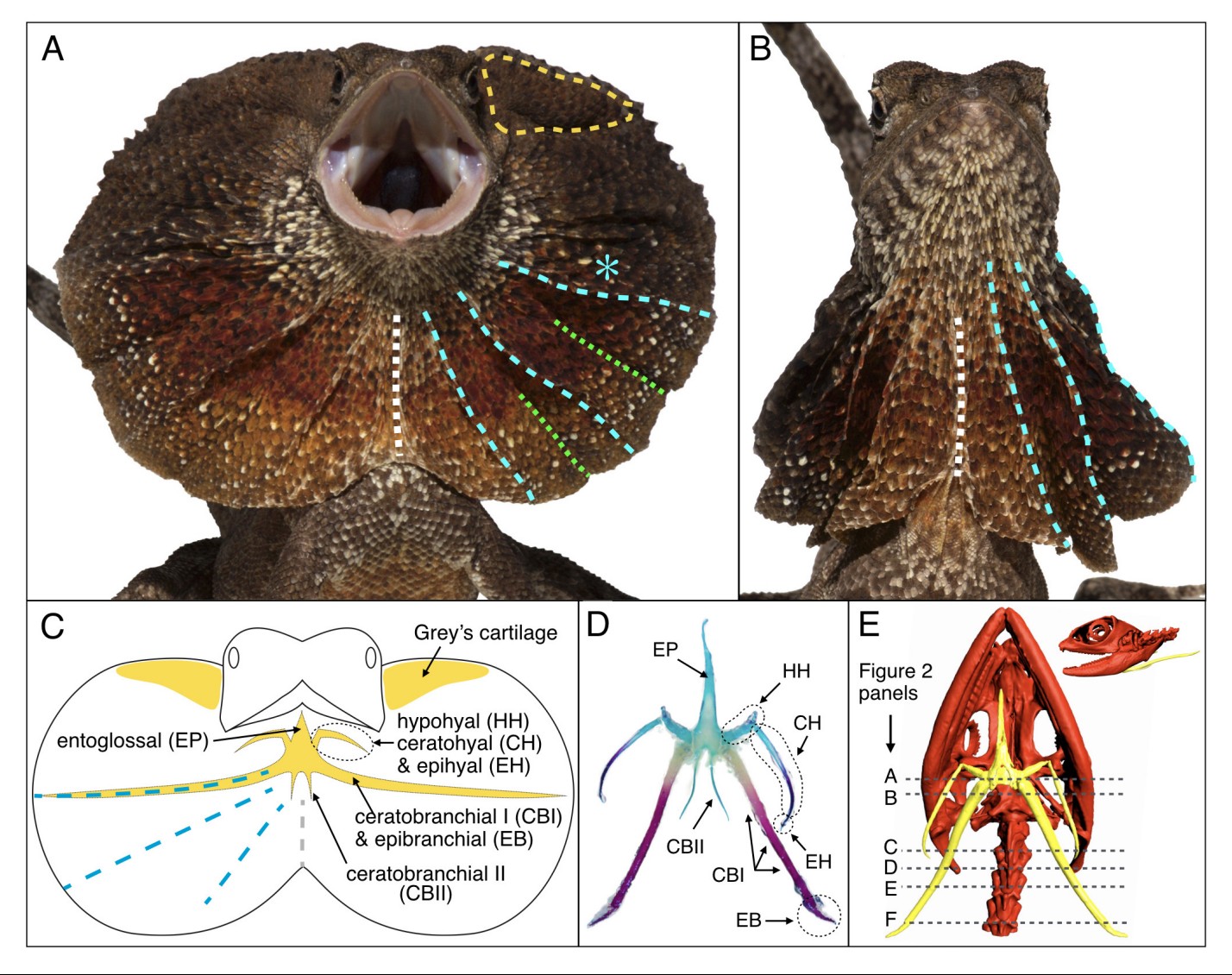

**Figure 1.** Morphology of the *Chlamydosaurus* frill and associated hyoid apparatus. (A) Erected and (B) folded frill of a juvenile (11 months old) frilled dragon. The left and right lobes of the frill each presents 3 pairs of convex ridges (cyan lines), two pairs of valley folds (green lines); the two lobes are joined at a central crease (white line). (C) The CBI + EB and 'Grey's cartilage' (yellow line in A) are embedded in the third convex ridge (cyan star in A) and the dorsal side of the frill, respectively. (D) Adult hyoid apparatus treated with alcian blue (staining cartilage) and alizarin red (staining bone). 'EP': entoglossal process, 'HH': hypohyal, 'CH': ceratohyal, 'EH': epihyal, 'CBI': ceratobranchial I, 'EB': epibranchial, 'CBII': ceratobranchial II. The central and anterior parts of the hyoid apparatus and first pair of horns (HH +CH + EH) originate from the second branchial arch (BA2), whereas the second (CBI + EB) and third (CBII) horns originate from the BA3 and BA4, respectively. (E) CT scan of an adult female (yellow, hyoid apparatus; red, skull and vertebrae); dashed lines indicate the approximate position of histological sections in *Figure 2*.
DOI: https://doi.org/10.7554/eLife.44455.003

# Results

## The morphology of the frill

The frill of *Chamydosaurus* is a large and sagitally-symmetric piece of skin attached to the neck and head. The left and right lobes of the frill are connected ventrally by a central crease (white line if *Figure 1A*). When erected, following the movements of the mandible, hyoid bones and Grey's cartilages, the frill forms a flat disc positioned in a transversal plane (*Figure 1A*). At rest, each lobe of the frill always pleats into three convex ridges and two concave folds, in addition to the central concave crease (*Figure 1B*). We confirmed by computed-tomography (CT) that the lower part of the frill is

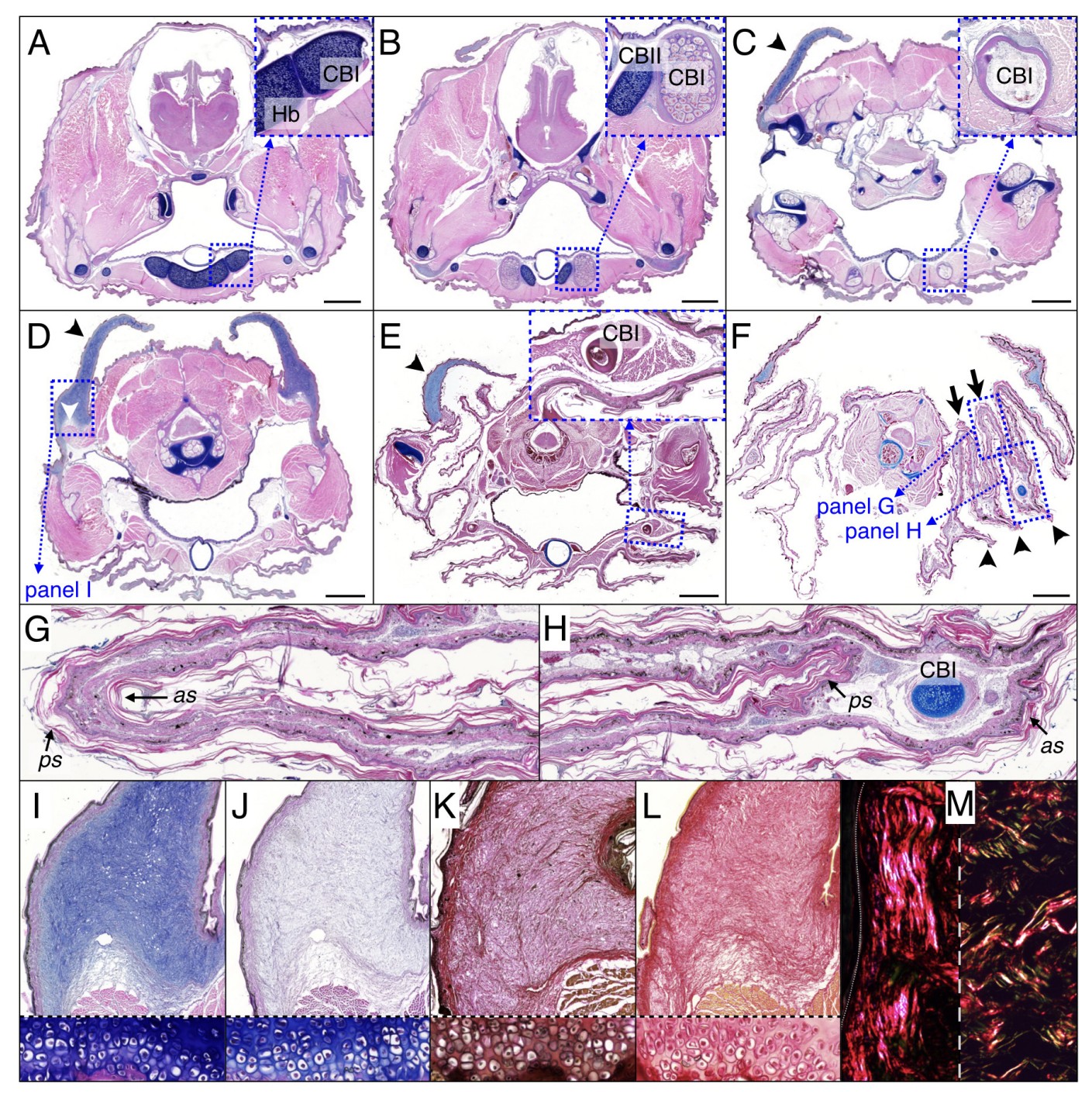

**Figure 2.** Histology of frill tissues and characterisation of the 'Grey's cartilage' in a 9 month-old *Chlamydosaurus.* Approximate positions of the transverse sections are indicated in *Figure 1E*. (**A**) The ceratobranchials I (CBI) are attached to the cartilaginous hyoid body (Hb) and (**B**) become ossified when they detached from the hyoid body, while the CBII remain cartilaginous. (**C,D**) The 'Grey's cartilage' (black arrowheads) is situated close to the tympanic membrane and its proximal end is attached to muscles (white arrowhead in D). (**E**) The CBI incorporates within the 3rd ridge. (**F**) The frill detaches from the throat and forms three convex ridges (arrowheads) and two concave folds (arrows). (**G,H**) Close-ups of anterior (as) and posterior (ps) skin sheets of panel F: *as* and *ps* have similar lengths in the two concave folds (**G**) whereas *as* is longer than *ps* in the three convex ridges (**H**). (**I,J**) Root of the Grey's cartilage (close-up of D): stained deep blue by alcian blue, hematoxylin and eosin (**I**), but dull-blue by alcian blue and MgCl₂ (**J**), which indicates the presence of weakly-sulphated GAGs. (**K,L**) The Grey's cartilage does not contain elastic fibres (otherwise it would be stained blue-black in K by the 'elastic stain kit'), but is rich in collagen fibres that are stained red by the 'elastic stain kit' (**K**) and by Sirius Red (**L**). Lower panels in I-L

*Figure 2 continued on next page*

*Figure 2 continued*

show cell-resolution closeups of intervertebral fibrocartilage. (**M**) Under polarised light, Sirius Red staining reveals a majority of thick fibres at the vicinity of the epidermis (left panel) and a mix of thin and thick fibres deeper in the tissue (right panel). Scale bars = 2 mm.

DOI: https://doi.org/10.7554/eLife.44455.004

supported by hypertrophied CBI bones (*Beddard, 1905*). Unlike in other agamid lizards, such as *Pogona* spp (*Throckmorton et al., 1985*), where the CBIs are fully enclosed in the throat and do not extend further than the end of the lower jaw, the frilled dragon's straight and much elongated CBIs have most of their length positioned into the frill (*Figure 1C*). Skeleton staining and paraffin sections indicate that the frilled dragon CBI bones are ossified whereas other parts of the hyoid apparatus remain cartilageneous (*Figures 1D* and *2A–F*). Anteriorly, the CBIs are attached to the posterior part of the hyoid body within the throat (*Figure 2A*) and are surrounded by muscles. More distally, the CBIs separate from the throat to become incorporated into the third ridge of each lobe of the frill (*Figure 2E–F*). In its free part, the frill consists of two sheets of skin linked by loose connective tissue (*Figure 2G,H*), with the anterior sheet being longer than the posterior sheet at the positions of the convex ridges (*Figure 2F,H*).

The Grey's cartilages (black arrowhead in *Figure 2C–E*) connect the dorsal part of the frill to each side of the head at the vicinity of the tympanic membrane. The proximal end of the Grey's cartilage is strongly attached to muscles (*Figure 2D*), the *digastric*, the *attolens chlamydis* and the *adductor chlamydis*, that allow for the movement of the upper part of the frill (*De Vis, 1883*). We used staining techniques to investigate further the nature of the Grey's cartilage, described previously as 'fibro-cartilagenous' (*De Vis, 1883*). As Alcian blue strongly stains the structure in the absence (*Figure 2C–E,I*), but not in the presence (*Figure 2J*), of $MgCl_2$, it is likely to contain weakly-sulphated glycosaminoglycans (GAGs) rather than keratan sulphates (*Bancroft, 2002*; *Scott and Dorling, 1965*); the latter are characteristic components of true cartilage. Elastic staining does not reveal elastic fibres (*Figure 2K*), while 'Sirius Red' staining indicates the presence of collagen fibres (*Figure 2L*). As, under polarised light, thick and thin collagen fibres appear orange-red and green-yellow, respectively (*Rich and Whittaker, 2005*), we could infer that fibres are thick at the surface of the 'cartilage' (near the epidermis, *Figure 2M*, left panel) whereas they are heterogeneous in size deeper in the structure (*Figure 2M*, right panel).

## Morphogenesis of the *Chlamydosaurus* neck frill

We investigated the dragon's frill morphogenesis across pre-hatching development. Around embryonic day 23 (E23 = 23 days post-oviposition), the early frill is visible as a swollen skin outgrowth in the ventral portion of the neck above the heart cavity and shoulders (*Figure 3A*). Between E24 and E30, the frill splits into right and left lobes, the central ventral crease becomes visible, and the latero-dorsal side of the outgrowth reaches the tympanic membrane (*Figure 3B*). At the end of that period (*Figure 3C*), two ridges become clearly visible on the anterior surface of each lobe while the latero-dorsal part grows towards the back of the head, beyond the tympanic membrane. Between E30 and E40, a third ridge forms (*Figure 3D*). At E45, the basic morphology of the frill is established: the anterior side of each lobe has grown substantially and exhibits three expanded anterior ridges while the lateral side of the frill now extends well beyond the tympanic membrane (*Figure 3E*).

Our dissections of embryos at earlier stages indicate that the skin outgrowth generating the early development of the frill in *Chlamydosaurus* is present at E15 but is hidden behind the heart cavity. We investigate below this early outgrowth in the context of the branchial arches (BA) development.

## Incomplete fusion of BA2 allows the development of a neck skin fold in Episquamata and of the frill in *Chlamydosaurus*

Around E6-E7, the four first BAs (BA1 to BA4) are visible (*Figure 4A*) on *Chlamydosaurus* embryos while the BA6 is hidden behind the developing heart and is only discernible on parasagittal sections (*Figure 4B*). In amniotes, the posterior part of the BA2 (hyoid arch) has been called an 'embryonic opercular flap' because it grows, expands caudally and covers the posterior BAs. Eventually, the BA2 fuses to the cardiac eminence, causing the internalisation of the BAs 3 to 6 (*Richardson et al., 2012*). In the frilled dragon, the posterior BAs internalise around E11 (*Figure 4C,D*) and are not

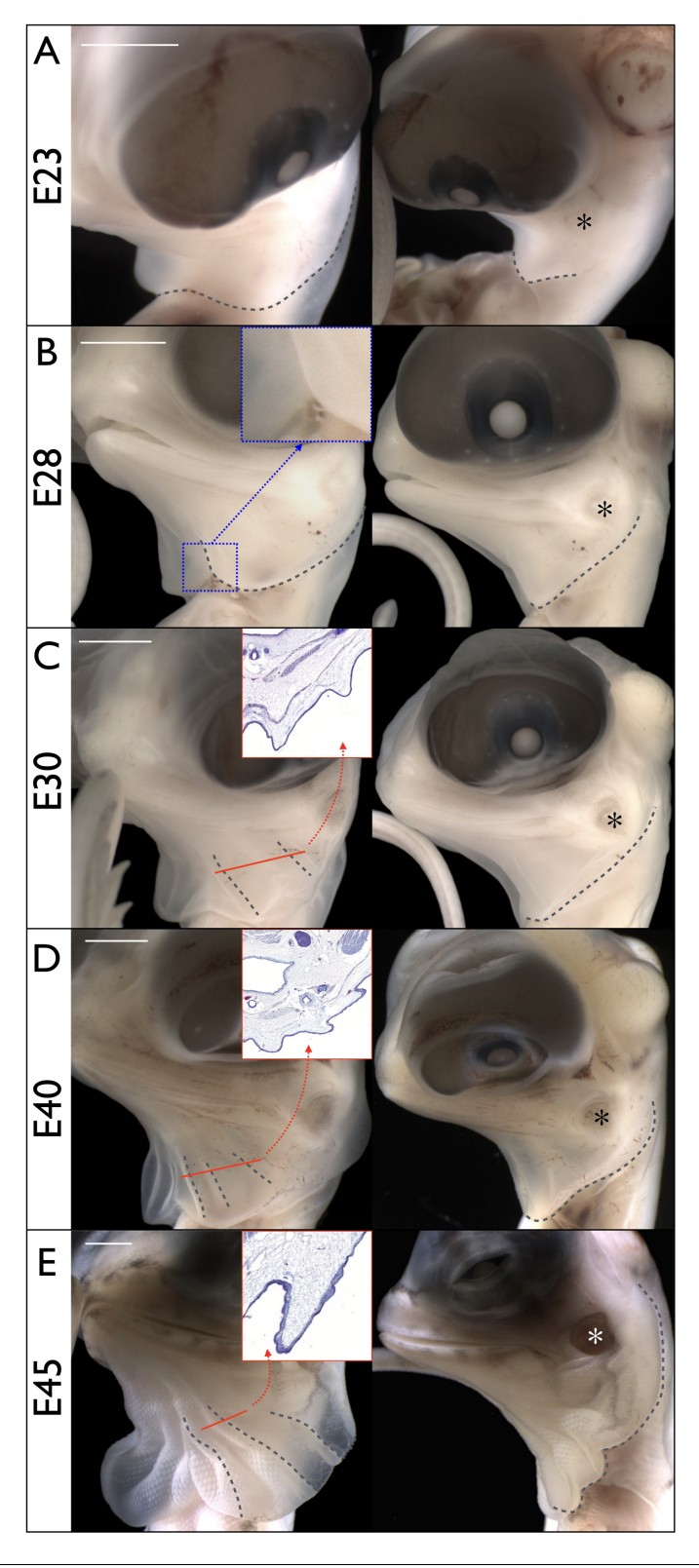

**Figure 3.** Morphogenesis of the *Chlamydosaurus* frill. Left and right columns are frontal and lateral views of the same embryo; E = days post-oviposition; grey dotted lines: frill outline and convex ridges. (**A**) At about E23, the developing frill is visible as a smooth skin outgrowth. (**B**) At E28, the central crease (inset) is clearly visible and the dorsal side of the frill extends to the tympanic membrane (asterisks in all right panels). (**C**) At E30, two convex pleats are visible on each lobe of the frill and the dorsal side extends beyond the tympanic membrane. (**D**) At E40, three pleats are visible. (**E**)
*Figure 3 continued on next page*

*Figure 3 continued*

At E45 the morphogenesis of the frill is mostly achieved. Although no pleat is visible in simple bright field microscopy at E28, episcopic microscopy indicates that low amplitude folding has already started at that stage. The insets in C, D, and E show the folding of the anterior surface of the frill at the position of the corresponding red line. Scale bars: 2 mm.

DOI: https://doi.org/10.7554/eLife.44455.005

discernible anymore at E15-E16 (*Figure 4E,F*). However, here we observe that part of the frilled dragon's BA2 (arrowhead in *Figure 4F and G*) does not fuse to the cardiac eminence, forming the early frill behind the heart (*Figure 4G*) before intensively growing (*Figure 4H,I*). Hence, we show

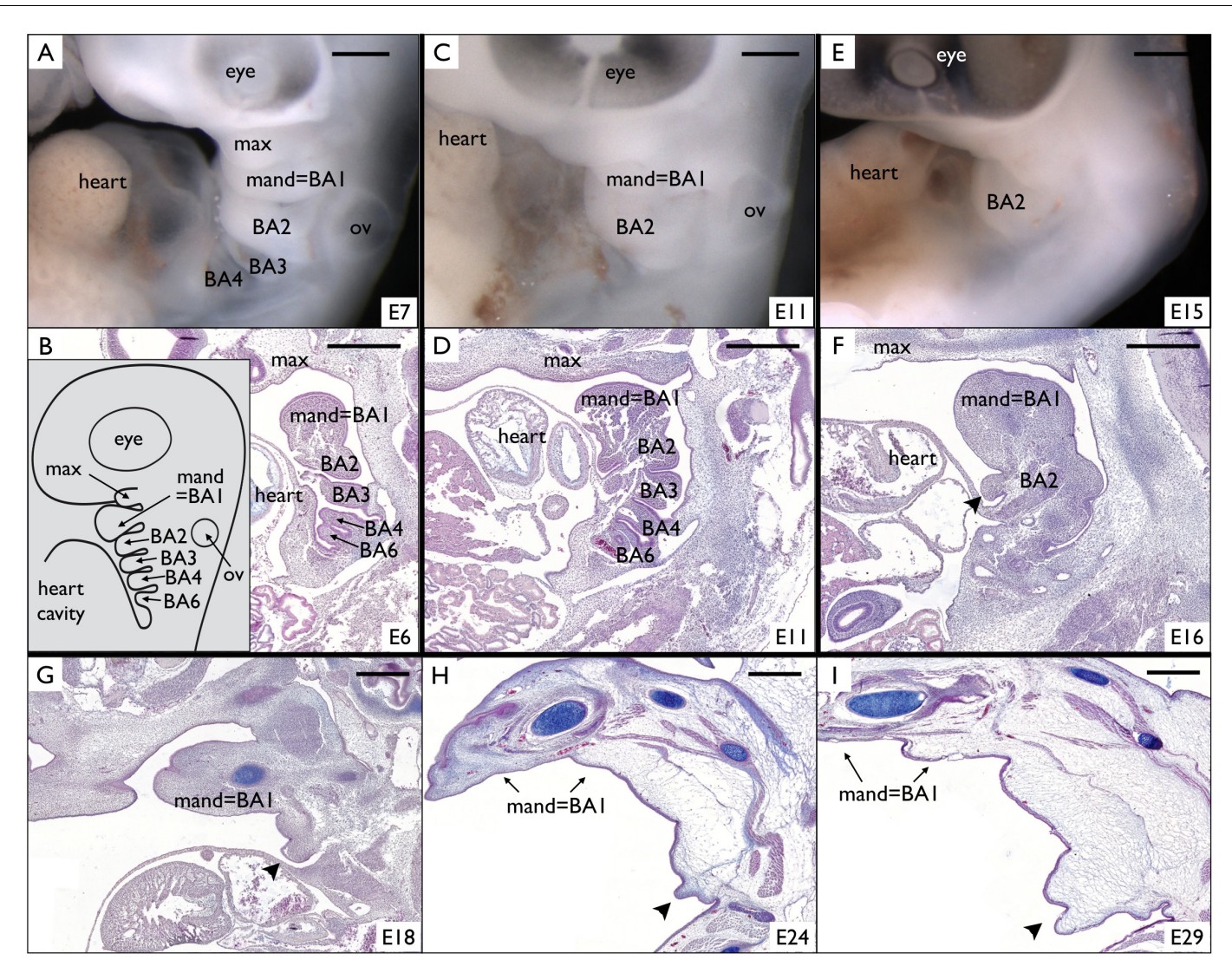

**Figure 4.** Incomplete fusion of BA2 allows the development of a frill in *Chlamydosaurus*. (A) At developmental stage 30 (based on *Wise et al., 2009*, that is E6-7 in *Chlamydosaurus*) the BA2, BA3, and BA4 are situated behind the hearth cavity and (B) the BA6 is only visible on parasagittal sections. Inset of B: schematic diagram of branchial arches organisation in land vertebrate embryos. (C,D) At stages 30–31 (about E11), the BA2 is still externally visible while the more posterior branchial arches are internalised. (E,F) At stage 32, (*i.e.*, about E15-16), a portion of BA2 (arrowhead) does not fuse with the cardiac eminence. (G,H,I) At later developmental stages (stages 32–33, 34, and 36, that is about E18, E24, and E29, respectively), the unfused portion of BA2 (arrowheads) intensively grows and generates the frill. 'BA1' to 'BA6'=branchial arches 1 to 6; 'heart'=heart cavity; 'max'=maxillary process; 'ov'=otic vesicle. Note that the first branchial arch (BA1) generates the mandibular process ('mand'). Scale bars = 500 µm.

DOI: https://doi.org/10.7554/eLife.44455.006

here that, in *Chlamydosaurus*, the frill originates from the outgrowth of the BA2 that failed to completely fuse with posterior BAs and the neck.

We then studied the internalisation of BAs in other species of squamates. We identify the presence of an incompletely fused BA2 for an extended period of embryonic development in all species investigated (*Figure 5*): the leopard gecko (*Eublepharis macularius*), the ocellated lizard (*Timon lepidus*), the veiled chameleon (*Chamaeleo calyptratus*), and the bearded dragon (*Pogona vitticeps*). The maintenance of the BA2 eminence throughout pre- and post-hatching development in the ocellated lizard and bearded dragon is responsible for the formation of a clearly visible neck fold (*Figure 5F,H* and *Figure 5—figure supplement 1A–F*). In the leopard gecko and veiled chameleon, the incompletely fused BA2 will remain visible up to at least developmental stages 34 (following developmental staging system of *Wise et al., 2009*) but will eventually fuse with the neck at later stages (*Figure 5—figure supplement 1G–L*), explaining that these species do not exhibit a conspicuous neck fold (*Figure 5E,G*).

Given that (i) the opercular flap (BA2) fuses with the cardiac eminence in non-squamate amniotes such as birds and mammals (at stages E6 and E10 for the chicken and the mouse, respectively), (ii) all Squamata lineages we investigated exhibit an unfused opercular flap for some period of their development, and (iii) many species of Episquamata exhibit a neck skin fold (cf. green lineages in *Figure 5*), the most parsimonious evolutionary scenario (*Figure 5*) is that the opercular flap forms in all amniotes but it's complete fusion was abolished at the origin of the Episquamata clade, that is after the divergence of the more ancestral lineages of Gekkota and Scinciformata. This event allowed the development of a neck skin fold in the ancestor of Episquamata, a morphological feature that was secondary lost (cf. red lineages in *Figure 5*) in chameleons, snakes and various legless lizards. The neck fold was then developed and modified into a spectacular erectile neck ruff during the evolution of the frilled dragon.

## The *Chlamydosaurus* frill ridges are generated by an elastic instability

The folding pattern of the dragon's frill is robust: the left and right lobes of the frill each pleats into three convex ridges and two concave folds (*Figure 1A,B*). In the two concave folds, the anterior and posterior skin sheets have similar lengths (*Figure 2F,G*) whereas the anterior skin sheet is substantially longer than the posterior skin sheet in the three convex ridges (*Figure 2F,H*). This morphology indicates that the convex ridges impose the frill to pleat at these positions, inevitably causing the concave folds to also occur.

To investigate the origin of the pre-folded pattern, we first tested whether they are generated by local increased proliferation. Indeed, if the anterior surface of the developing frill exhibited six lines of increased growth (three on the left lobe and three on the right lobe) superposed to the future location of the ridges, it would explain that the frill folds exactly there because the anterior skin sheet would become locally larger than the posterior skin sheet (*Figure 2F,H*). Such a pattern of localised growth could be controlled by a corresponding pattern of morphogen gradients generated by a Turing-like (reaction-diffusion) mechanism or by unknown positional information. Hence, we used the mitotic marker phospho-Histone H3 (pH3) to quantify proliferation across the frill while and after ridges are formed. Our analyses indicated (i) similar cell densities (of about 0.02 cell per $\mu m^2$) and proliferation during and after ridges formation, and (ii) no notable difference in proliferation at the location versus in between ridges (*Figure 6*). Although these proliferation analyses are limited by the low number of embryos at our disposal, the data generated does not hint at any obvious proliferation spatial patterning.

This leaves us with the possibility that the ridges of the dragon's frill are generated mechanically by frustrated homogeneous growth. Indeed, recent analyses have demonstrated the importance of mechanical instabilities in morphogenesis (*Nelson, 2016*). For example, uniform growth of the gut at a rate larger than that of the anchoring dorsal mesenteric sheet is sufficient to quantitatively explain the gut looping morphogenesis into the body cavity of vertebrates (*Savin et al., 2011*). Similarly, it has been suggested that the folding of the developing cerebral cortex in mammals is caused by expansion of the grey matter constrained by the more-slowly developing white matter (*Karzbrun et al., 2018*; *Richman et al., 1975*; *Tallinen et al., 2016*).

Using 3D measurements on frilled dragon embryos at various developmental stages, we observe that the linear dimension of the frill surface (square root of the area) increases approximately 1.3 fold relative to the length of the frill boundary attached to the neck (*Figure 7*). This observation is

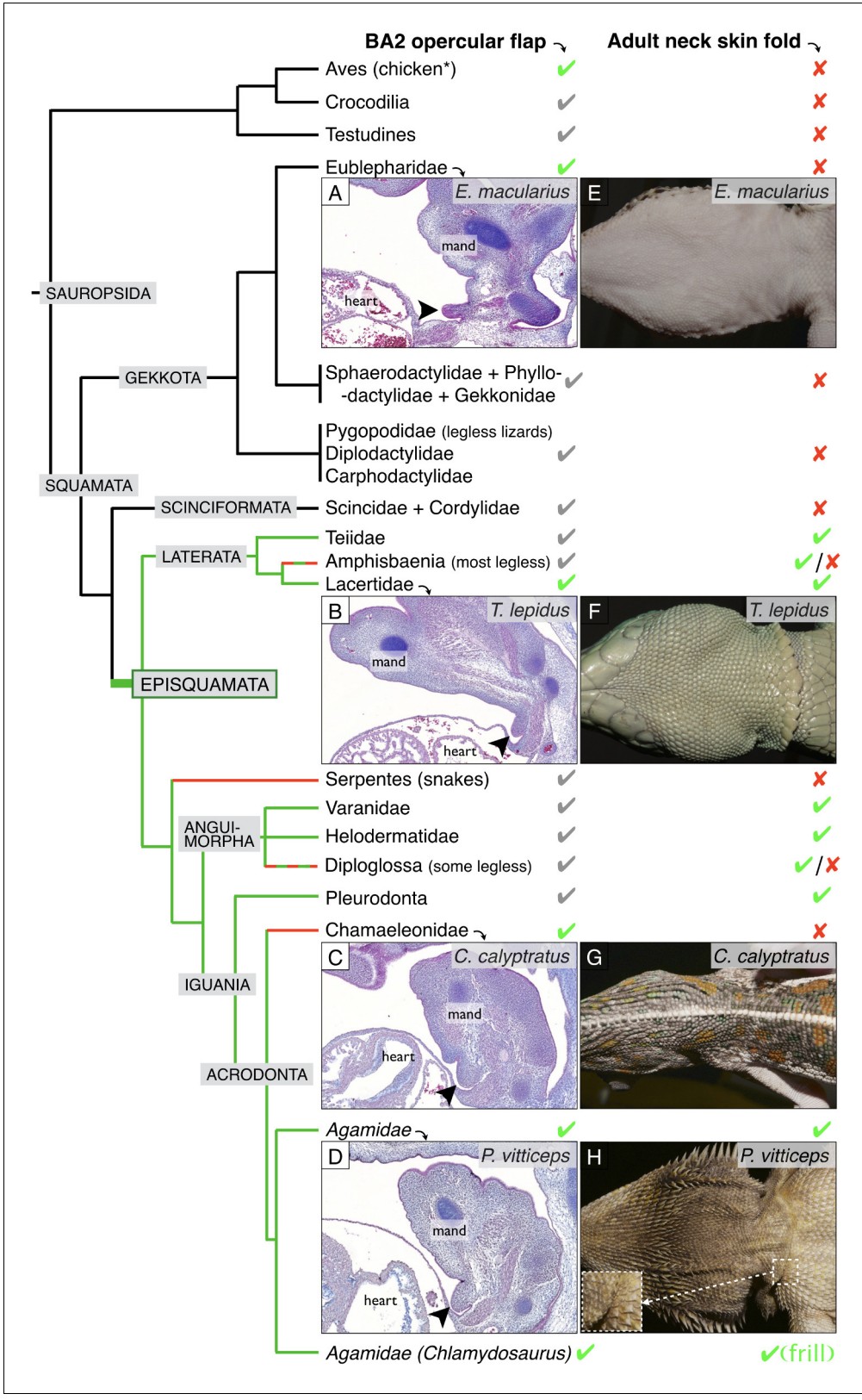

**Figure 5.** Incomplete fusion of BA2 occurred in the ancestor of Episquamata. Phylogenetic tree (based on *Zheng and Wiens, 2016*) of Sauropsida and sagittal sections revealing the presence of an embryonic opercular flap (arrowheads) in (**A**) *Eublepharis macularius*, leopard gecko (stage 32), (**B**) *Timon lepidus*, ocellated lizard (stage 33), (**C**) *Chamaeleo calyptratus*, veiled chameleon (stage 32), and (**D**) *Pogona vitticeps*, bearded dragon (**E10**). In adults, a neck skin fold is (**E**) absent in the leopard gecko, (**F**) present in the ocellated lizard, (**G**) absent in the veiled chameleon, and (**H**) present in the
*Figure 5 continued on next page*

*Figure 5 continued*

bearded dragon. As, to our knowledge, no species of Gekkota and Scinciformata exhibit a neck fold, the most parsimonious phylogenetic mapping is that the neck fold evolved in the ancestor of Episquamata (thick green branch) and was subsequently lost (red branches) in the morphologically highly-derived snakes (Serpentes), Chameleonidae, and various legless lizards. The green check marks indicate presence of a character confirmed by our direct observations, whereas grey marks indicate expectation on the basis of the parsimony argument. Red crosses indicate the absence of a character. 'Mand'=mandibular arch, 'heart'=heart cavity.

DOI: https://doi.org/10.7554/eLife.44455.007

The following figure supplement is available for figure 5:

**Figure supplement 1.** Development of the BA2 in various lineages of squamates.

DOI: https://doi.org/10.7554/eLife.44455.008

compatible with a physical (mechanical) morphogenesis process where homogeneous growth of the frill's anterior skin sheet, frustrated by a boundary condition (its attachment to the neck), generates an elastic instability that resolves into the formation of three anterior convex ridges on each lobe of the dragon's frill. In other words, contrary to the folding of the brain, which is geometrically constrained all over its surface, the frill might be folding during development because of a 'curtain-instability process', similar to the wrinkling of gravity-induced draping (*Cerda et al., 2004*; *Vandeparre et al., 2011*) where self-similar wrinkled patterns are generated under boundary confinement.

We then construct a more complex model to assess if the folding pattern is additionally influenced by the attachment of the frill's anterior skin sheet to the underlying loose connective tissue. Below, we measure and implement realistic tissue physical parameters and growth rates into numerical simulations and physical analog experiments to test whether such models can quantitatively predict the robust folding pattern of the *Chlamydosaurus* dragon's frill.

## Numerical simulations

Our 3D numerical growth model, based on a custom finite element method (see Supplementary Methods), assumes that the two lobes of the frill grow independently and are symmetrical with respect to the central crease. We first use a thin semi-cylindrical geometry of thickness $T$ and diameter $L$ (*Figure 8A*) as a simplified model of the embryonic frill anterior surface. We model the frustrated growth of the sheet (*i.e.*, the smaller growth rate of the edge attached to the neck relative to the growth rate of the frill) as follows: the length $L$ of the sheet's straight edge is maintained constant while tangentially growing the rest of the structure. This simple model indicates that the number of ridges increases with the amount of expansion $g$ (*Figure 8B* and *Figure 8—figure supplement 1*), while it decreases with increasing relative thickness ($T/L$) of the sheet (*Figure 8C* and *Figure 8—figure supplement 1*). To generate a more realistic geometry of the anterior surface of the frill, we apply a curved neck boundary and the presence of a central crease (*Figure 8D*). Using this geometrical configuration and a quasi-static approach, our simulations recapitulate the transition from two to three ridges observed during embryonic development: for $T/L = 0.014$ (*i.e.*, the ratio between the average values of skin thickness and the neck boundary length measured on real embryos; *Figure 7*), each lobe of the frill exhibits two ridges when $g$ reaches about 1.07 and three ridges when it exceeds 1.15 (*Figure 8E* and *Video 1*).

To further test the single sheet model, we performed a physical analog experiment using a thin semi-cylindrical sheet of polydimethylsiloxane (PDMS) gel with a $T/L$ value of 0.01 (*Figure 9A*) and constructed an equivalent computer model (*Figure 9B*). We fixed the PDMS sheet at its straight edge and used hexane to make it swell by about 30%. The resulting geometry (*Figure 9C*) is very similar to that obtained with our numerical simulations (*Figure 9D*): the swollen PDMS sheet exhibits three wrinkles with a wavelength that increases towards the free edge.

Next, to investigate the influence of the substrate (loose connective tissue) on the resulting morphology, we generated a more complex model (*Figure 10A*) derived from the actual frill geometry obtained from a high-resolution episcopic microscopy (HREM) 3D reconstruction at E23, that is a stage where the frill appears as a swollen skin outgrowth with no ridge (*Figure 3A*). As the thickness of the anterior skin sheet exhibits nearly constant values during the period of ridges formation (*Figure 7C*), we used its average value of 47 μm. By measuring 3D morphological features of the dragon's frill between E23 (*i.e.*, when the anterior skin is smooth) and E32 (*i.e.*, when the third ridge

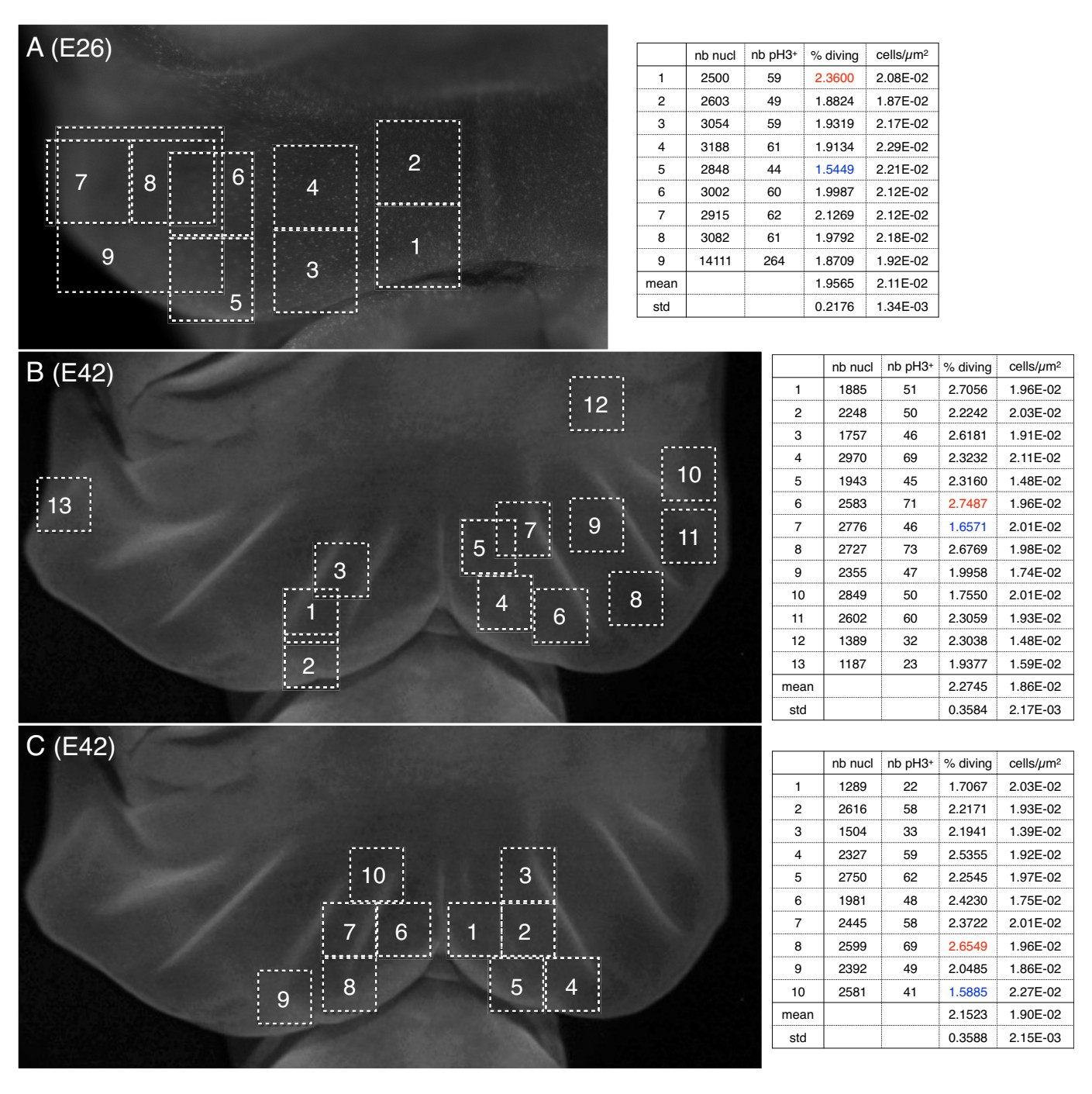

**Figure 6.** Cell density and proportion of dividing cells. Quantification of nuclei and of pH3-labelled cells (phospho-histone H3, a mitotic marker) (A) before and (B,C) during the formation of ridges on the *Chalmydosaurus* embryonic frill. White squares: positions of the surface of tissue investigated by confocal microscopy; z-stacks were generated and the number of nuclei and of pH3-labelled nuclei were counted (values are indicated in the tables on the right; numbers in red and blue are highest and lowest values, respectively).
DOI: https://doi.org/10.7554/eLife.44455.009

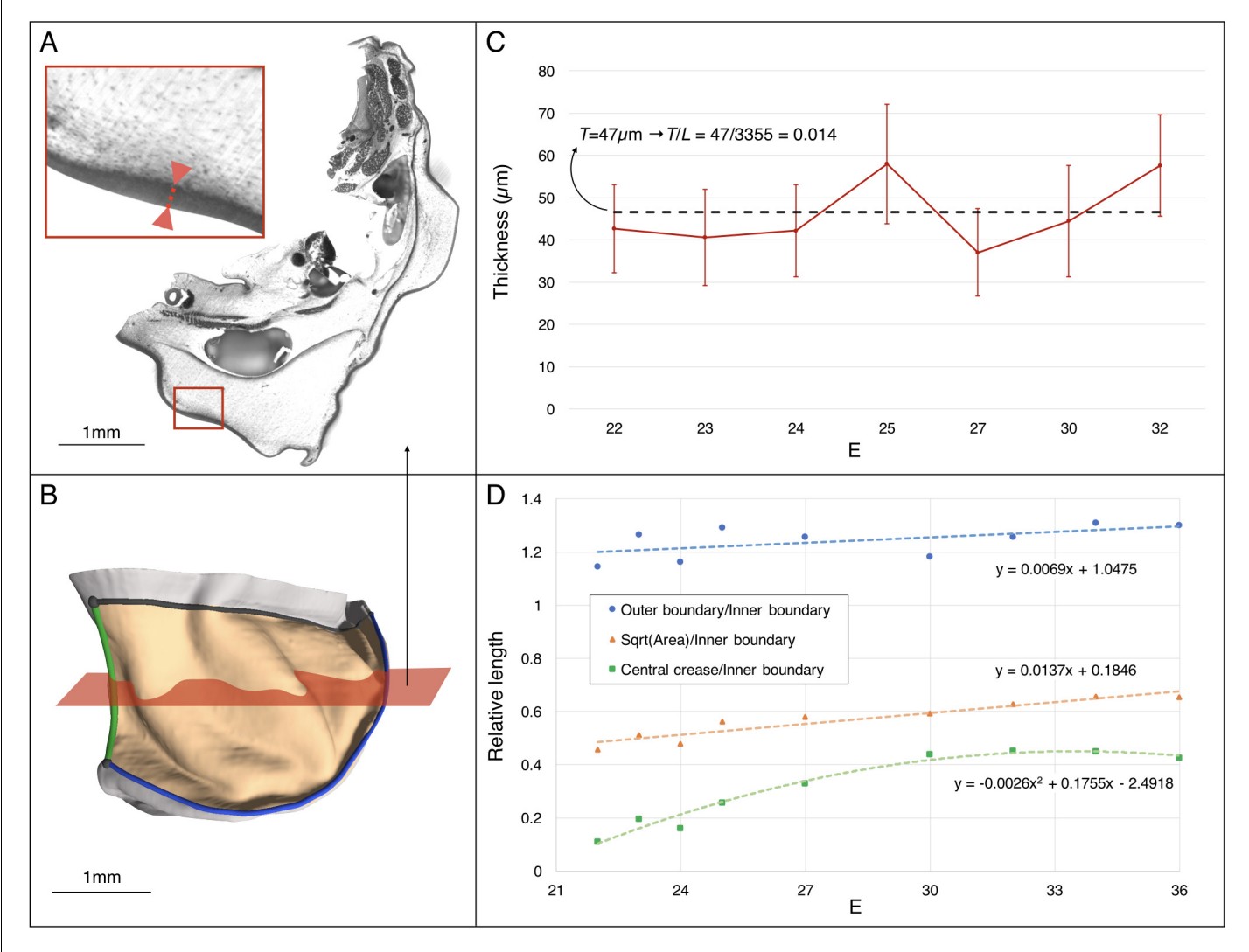

**Figure 7.** Estimating the changes in thickness and boundary conditions during *Chalmydosaurus* embryonic frill development. (**A**) Eosin staining of an HREM slice at E36 shows the sharp difference in contrast between the frill's skin (red double arrow in inset) and the underlying loose connective tissue. (**B**) 3D reconstruction of the frill obtained by stacking HREM slices. The central crease is marked in green, the outer boundary in blue, the inner boundary (attached to the neck) in black, and the area of the frill in light orange. The approximate position of the slice in panel A is indicated by the red plane. (**C**) Average skin thickness measured at different developmental stages (horizontal axis: E = days post-oviposition); error bars represent one standard deviation intervals (skin thickness varies at different places of the frill) for the embryo at the corresponding stage. The thickness value used in our simulations is represented by the average (dashed line) among the seven measurements. (**D**) Three-dimensional measurements among different parts of the frill between E22 and E36 (one embryo per stage) indicate a linear surface growth g(s) ≈ 1.3, a central crease growth g(c) ≈ 2.2, and an outer free edge growth g(o) ≈ 1.0 during the developmental period when the ridges form (E23-32). Note that the growth values used at different developmental stages during our quasi-static simulations (*Figure 10C*) are derived from the linear fit to the linear surface data and the quadratic fit to the central crease data shown here.

DOI: https://doi.org/10.7554/eLife.44455.010

is clearly formed), we estimate: (i) the linear surface growth, that is the expansion perpendicular to the surface normal of the frill to be g(s) ≈ 1.3; (ii) the central crease growth (in the direction C in *Figure 10A*) to be g(c) ≈ 2.2, and (iii) the outer free edge growth of the frill to be g(o) ≈ 1.0. All these values are relative to the length of the neck boundary (*Figure 7D*) and are used to grow our numerical model (*Figure 10*). Note that we assume the simulated materials to be nearly incompressible (Poisson's ratio v = 0.45), as suggested for biological soft tissues such as skin (*Choi and Zheng, 2005*; *Hendriks et al., 2003*; *Khatyr et al., 2004*). Our simulations show that the frill's surface

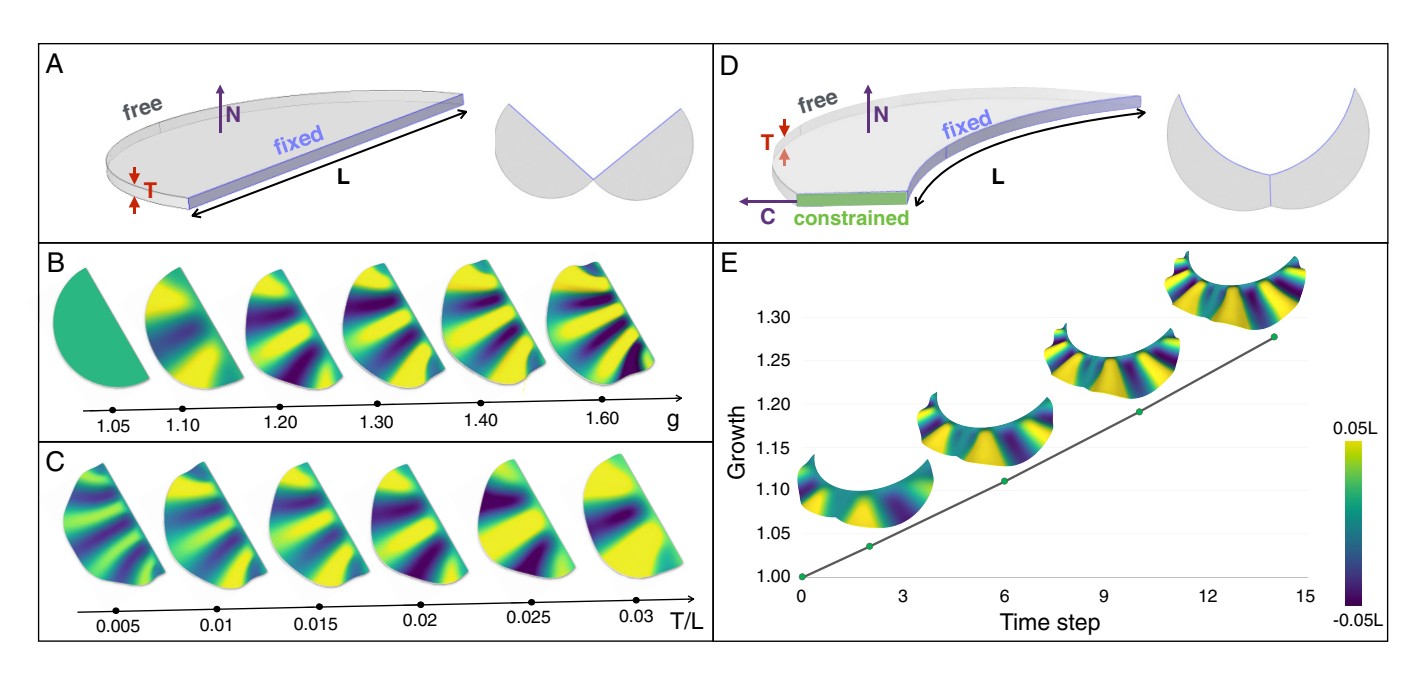

**Figure 8.** An elastic instability generates ridges during homogeneous growth of a thin sheet computational model. (**A**) Semi-cylindrical model of diameter L and thickness T with fixed inner boundary and free outer boundary. (**B**) For a constant ratio T/L (=0.02), the number of ridges generated by uniform growth (g) increases with increasing value of g; (**C**) For a constant amount of growth (here, g = 1.2), the number of ridges is negatively correlated with the ratio T/L. (**D**) A more realistic geometry of the frill with a fixed circular inner boundary (representing the attachment to the neck) and a central crease. (**E**) Assuming that the central crease and the linear surface grow at the same rate, a quasi-static approach to homogeneous growth of the model shown in D generates two then three ridges; T/L = 0.014 and Poisson's ratio ν = 0.45.
DOI: https://doi.org/10.7554/eLife.44455.011

The following figure supplement is available for figure 8:

**Figure supplement 1.** Influence of growth and of sheet thickness on the number of ridges.
DOI: https://doi.org/10.7554/eLife.44455.012

indeed wrinkles and that the number of convex ridges decreases with the skin-to-substrate stiffness ratio (*Figure 10B*), as expected from the formula $\lambda = 2\pi T (\mu_K/3\mu_s)^{1/3}$ (*Kim et al., 2011*; *Wang et al., 2016*) where $\lambda$ is the wrinkling wavelength of a compressed stiff film (here, the skin of thickness *T* and shear modulus $\mu_k$) on a flat soft substrate (here, the loose connective tissue of shear modulus $\mu_s$). As our simulations are performed on a curved geometry with fixed inner edge (attachment to the neck), they generate wrinkles that are less regular than those of flat film-substrate systems described by the above equation. For $\mu_k/\mu_s$ values $\leq 50$, the wavelength predicted by the equation is larger but similar (within two standard deviations) to the mean wavelength generated in the corresponding simulations. The discrepancy increases substantially for larger $\mu_k/\mu_s$ because the fixed inner edge in the simulated model increasingly prevents the wrinkles to 'repel' each others.

Note also that actual growth and elastic deformations are superposed in our measurements on embryos. The elastic contribution is small: at steady state, we measure a linear surface relative growth of 1.27 for an actual growth of 1.3 (using a stiffness ratio of 100).

Crucially, as our model generates three ridges for skin-to-substrate stiffness ratios in the range 65–1000 for g(s)=1.3, we need to evaluate if the actual value for the developing frill (*i.e.*, the ratio between the stiffnesses of the frill's surface skin sheet and of the underlying loose connective tissue) is within that range. Elastic moduli reported throughout the literature indicate several orders of magnitude variation in stiffness among human tissues (*Cox and Erler, 2011*). Hence, we performed depth-dependent nanoindentation measurements (*Figure 11* and Supplementary Methods) on fresh unfixed *Chlamydosaurus* embryos and inferred that the frill skin elastic modulus increases from 11'440 ± 200 Pa at E26 to 36'880 ± 20 Pa at E45 (*i.e.*, well in the 8–540 kPa range reported for

human skin [*Iivarinen et al., 2014*]), whereas the loose connective tissue modulus remains constant: 220 ± 7 Pa at E26 and 228 ± 14 Pa at E45. These numbers yield an increase of skin-to-substrate stiffness ratio from 52 to 162 during the development of the ridges. Note that a stiffness ratio of 52 at E26, that is smaller than the lower bound of the range 65–1000 required to generate three ridges, is not problematic because, at that stage, the frill grew only by $g(s)=1.1$ instead of $g(s)=1.3$. Implementing this smaller growth value (and a corresponding observed $g(c)=1.5$; *Figure 10C*) in our simulation model generates two ridges, as observed in real embryos. Finally, instead of applying the whole growth at once during numerical simulations, and checking the steady-state result, we also performed a quasi-static approach to progressively grow the HREM-derived realistic 3D model of the frill; skin-to-substrate stiffness ratio was set to 100 for convenience. These new simulations recapitulate the transition from two to three ridges observed during actual frill morphogenesis (*Figure 10C*, *Video 2*).

In principle, these semi-quantitative validations of our model could be reinforced by the comparisons of the amplitudes of the folds produced by the model with those observed on the developing embryos. The amplitudes of the middle ridge on the HREM reconstructions (104 µm at the end of the second ridge formation and 198 µm when the third ridge is formed) is larger than the corresponding amplitudes obtained with the numerical model (46 µm and 97 µm, respectively). This discrepancy remains even after normalisation: the corresponding relative amplitudes are 3.3% and 5.4% of the inner boundary (*i.e.*, the attachment to the neck) in the HREM reconstructions *versus* 1.7% and 3.6% in the simulations. The significance of these differences is difficult to evaluate because the simulations are performed using elasticity parameters evaluated on fresh tissues, whereas 3D measurements were done on HREM reconstructions after fixation and dehydration of the samples. As tissue shrinkage due to dehydration can dissimilarly affect different tissues, the relative amplitudes measured on the fixed samples might not accurately reflect their real values. In addition, we use a strictly elastic model whereas the actual embryonic tissues might experience flow. A full quantitative analysis of folding in the frilled lizard would require live-imaging (*e.g.*, with light-sheet microscopy), hence, the development of two techniques currently not available in reptiles: *ex-ovo* incubation (for imaging) and transgenesis (for fluorescent labelling of living tissues). Similarly, a better access and availability of *Chlamydosaurus* embryos would allow to perform controlled local tissue cutting experiments on multiple locations of the frill to measure the distribution of stress prior and during the development of the folds.

## Discussion

The emblematic erectile ruff of the frilled dragon is a large and sagitally-symmetric piece of skin attached to the neck and the head. At rest, the frill pleats into three convex ridges and two concave folds while the animal can spread this structure by the coordinated movements of hyoid-derived hypertrophied CBI bone (incorporated in the most dorsal ridge) and the so-called 'Grey's cartilage' that we identify not to be *bona fide* cartilage.

Here, we identify an ancient evolutionary developmental event that paved the way to the much more recent evolution of the spectacular *Chlamydosaurus* frill. Indeed, by comparing the embryonic development of representatives of the Squamata lineage, we suggest that the ancestor of Episquamata (*Figure 5*) lost the ability to completely fuse the hyoid branchial arch (BA2) with the cardiac eminence and posterior BAs, allowing for the transformation of this 'embryonic opercular flap' into a conspicuous neck fold. The latter was subsequently lost in chameleons, snakes as well as various legless lizards, while it hypertrophied in *Chlamydosaurus*.

Second, by producing and analysing embryonic series of frill dragons, we show that wrinkles form in the developing frill's anterior skin, establishing a pattern of three convex ridges that, later in development, allow the structure to robustly fold when rested along the animal's neck. Third, using histological data, analysis of proliferation, physical analogs and computational models, we suggest that the convex ridges are generated by an elastic instability rather than by local increased proliferation patterned by signalling morphogen gradients or positional information. Indeed, we show that homogeneous growth of the embryonic frill's anterior surface is sufficient to robustly produce on each lobe of the frill, first two then three convex ridges when the frill's growth is frustrated by its attachment to the neck. Finally, numerical simulations, implementing (i) a more realistic morphology (inferred from HREM 3D reconstructions) of the embryonic frill, incorporating the shape of it's skin

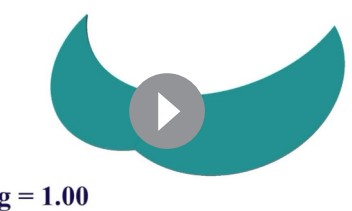

$g = 1.00$

**Video 1.** Quasi-static linear growth of a thin sheet simulation model of uniform thickness T with a curved inner boundary of length L (see *Figure 8D*) produces first two and then three ridges. Here we assume that the growth ($g = 1.28$ at the end of the simulation) experienced by the central crease and the surface are identical. Other model parameters are: T/L = 0.014 and Poisson's ratio ν = 0.45.

DOI: https://doi.org/10.7554/eLife.44455.013

and connective tissue substrate, (ii) measured mechanical parameters of *Chlamydosaurus* embryonic tissues, and (iii) a realistic growth model derived from embryonic series, indicate that the development of two ridges, and the later transition to three ridges, can be explained by a mechanical process that does not require any pre-patterning.

## Materials and methods

### Animals

Frilled dragons breed in the wild during the wet season from November to December (*Harlow and Shine, 1999*; *Shine and Lambeck, 1989*). In captivity, we obtained a mean number of 12.8 eggs per year per female. The incubation of frilled dragon's eggs is about 90 days at 29.5° C, that is substantially longer than in its close relative bearded dragon (*Pogona vitticeps*, 60 days). Maintenance of, and experiments on animals were approved by the Geneva Canton ethical regulation authority (authorisations GE/82/14, GE/73/16, and GE/27/19) and performed according to Swiss law. These guidelines meet international standards.

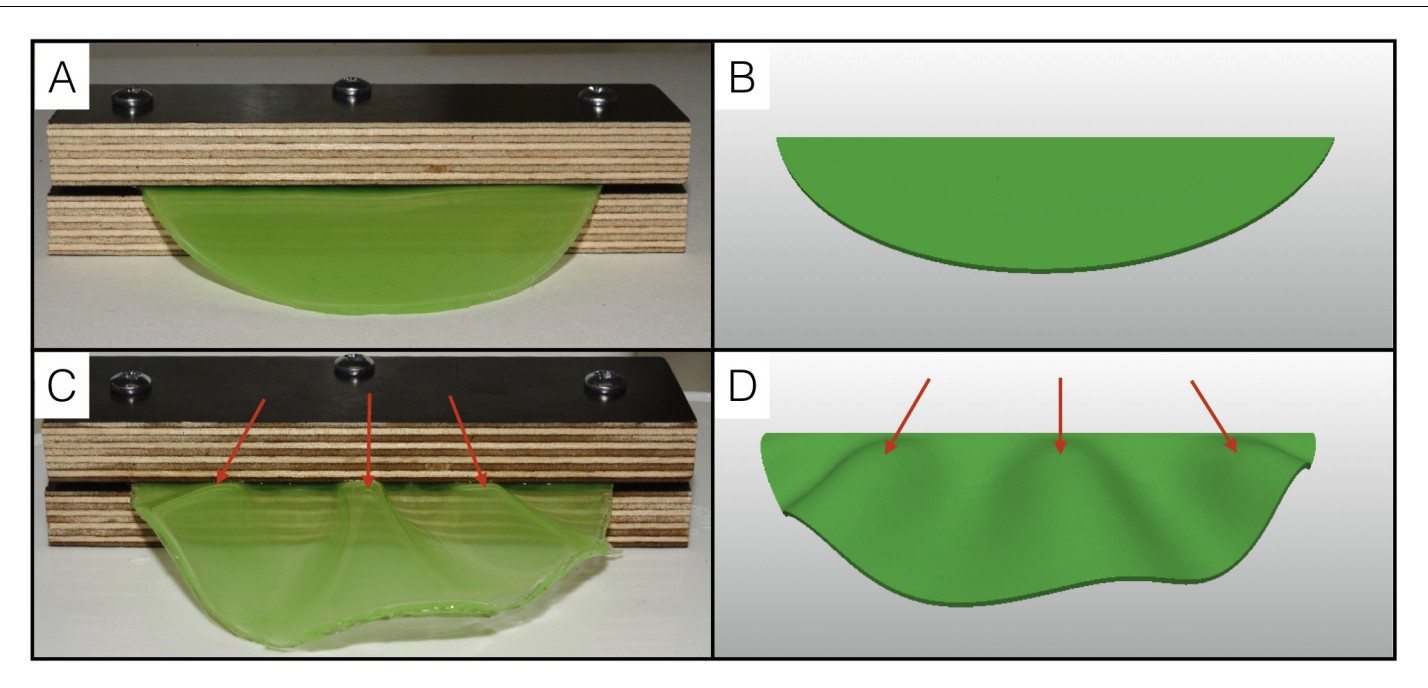

**Figure 9.** Physical analog experiment confirms numerical simulations. (**A**) A PDMS sheet of thickness T = 0.94 mm (measured at 10 positions, standard deviation = 0.05 mm) and inner boundary length L = 84.62 mm is fixed between two pieces of wood to mimic the neck boundary of the frill. (**B**) Tetrahedral mesh model with a T/L ratio (0.01) and a Poisson's ratio (ν = 0.45) similar to those of the PDMS analog shown in A. (**C**) After immersion in hexane (causing the swelling of the gel by a factor of 30%), the PDMS sheet forms three pleats as in (**D**) where the simulation mesh is grown a similar amount ($g = 1.3$) in all directions.

DOI: https://doi.org/10.7554/eLife.44455.014

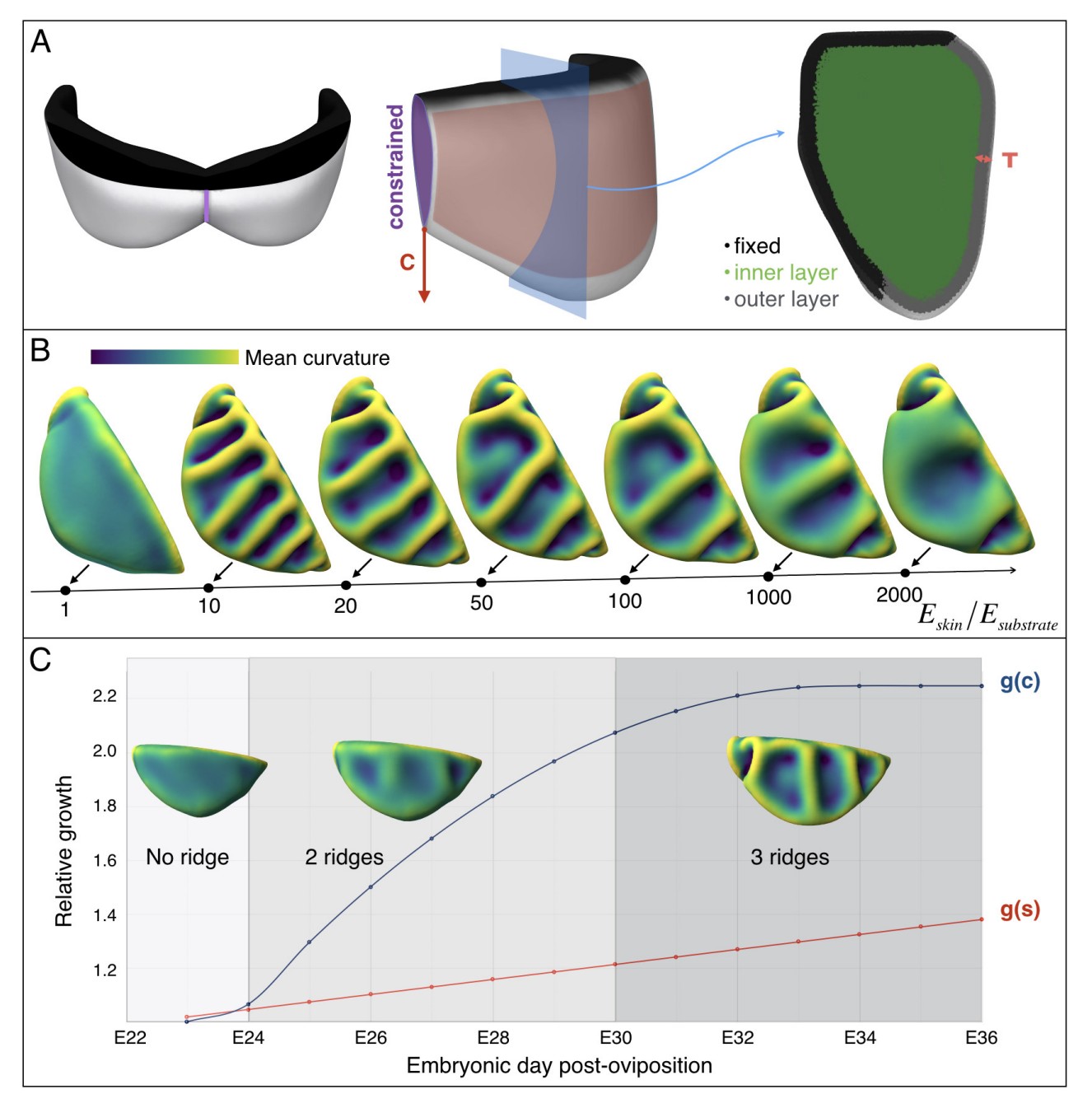

**Figure 10.** Multi-layer computational simulations of ridges formation using realistic embryonic frill geometry. (**A**) Computational model of the embryonic frill derived from measurements on a HREM 3D reconstruction of a E23 embryo's frill. The left, central and right panels show the model in dorsal, lateral and section views, respectively. The fixed inner boundary (attachment to the neck), central crease, and underlying loose connective tissue are indicated in black, purple, and green, respectively. Tangential growth is applied on the surface of the frill (anterior layer of skin indicated in pink and of thickness T = 47 μm) whereas the central crease is constrained to a plane and grows in the direction C. (**B**). Using this geometry and fixed relative growth (linear surface outer sheet growth g(s)=1.3 and central crease growth g(c)=2.2, both relative to the length of the neck boundary), we show that the number of ridges is inversely correlated with the skin-to-substrate stiffness ratio. Other simulation parameters are $v_{skin} = v_{substrate} = 0.45$. (**C**) Using a stiffness ratio $E_{skin}/E_{substrate}=100$, with dynamics of frill surface growth (g(s), red curve) and of central crease growth (g(c), blue curve) derived from the actual morphogenesis of the frill in *Chlamydosaurus* embryos (*Figure 7D*), simulations recapitulate the transition from two to three ridges observed in real embryos. Because of their low amplitude, the two ridges present in embryos at stages E24-E28 are difficult to identify in bright field microscopy (*Figure 3*) but are easily detected by episcopic microscopy.

DOI: https://doi.org/10.7554/eLife.44455.015

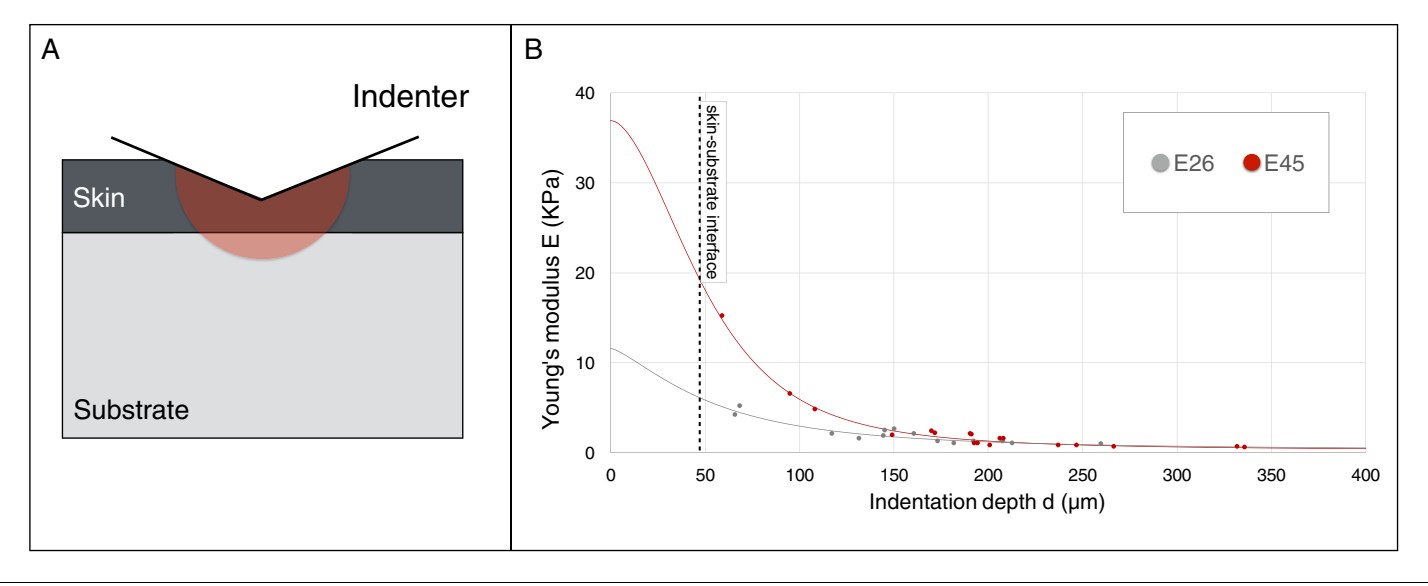

**Figure 11.** Young's modulus estimation using a micro-indenter. (**A**) Schematic representation of the hemisphere of influence of the micro-indenter. (**B**) Estimated Young's modulus E for skin thickness T = 47 μm, at two developmental stages E26 and E45 (one embryo each), as a function of indentation depth $d$. The values of $E_{skin}$ and $E_{substrate}$ were estimated using an approach described in the Material and methods.
DOI: https://doi.org/10.7554/eLife.44455.016

## Computed-tomography of skeletal elements

Computed-tomography scans were performed with a Skyscan-1076 microCT at a resolution of 35 μm (source: 55 kV, 179 μA). Three-dimensional iso-surfaces were created using the Imaris software (Bitplane, Zurich, Switzerland).

## Skeleton staining

Skinned heads were dehydrated and stained in 0.03% alcian blue in 80% EtOH and 20% acetic acid. The samples were rehydrated and stained in 0.01% alizarin red in 1% KOH. The hyoid apparatus was dissected and pictures taken with a Nikon D700 camera.

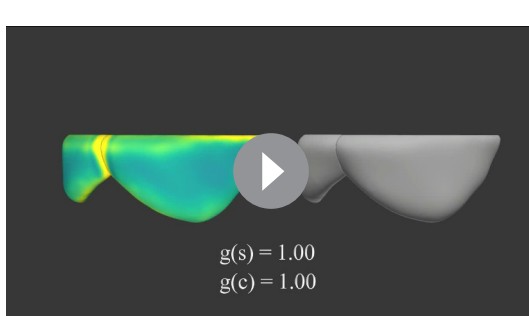

**Video 2.** Quasi-static growth of a realistic HREM-derived frill geometry (see *Figure 10A*), incorporating the frill's skin and the underlying softer connective tissue ($E_{skin}/E_{substrate}$ = 100), generates the transition from two to three ridges. Growth values g(s) and g(c) (indicated in the movie) are derived from the linear fit to the surface data and the quadratic fit to the central crease data, respectively (*Figure 7D*).
DOI: https://doi.org/10.7554/eLife.44455.017

## Histology

Tissue samples were fixed overnight in 4% PFA, rinsed in 1x phosphate-buffered saline (PBS). Post-embryonic samples were decalcified in Osteosoft (Merck, 1017281000). Samples were dehydrated before paraffin embedding and sectioned at 7 μm. For hematoxylin/eosin/alcian blue staining, sections were initially treated in a solution of 1% Alcian blue in 3% acetic acid before classical hematoxylin/eosin staining. For 'critical electrolyte concentration' applications (*Scott and Dorling, 1965*), the initial treatment was replaced by 0.05% alcian blue and 0.6M $MgCl_2$ in 0.2M acetate buffer. Sirius Red staining were performed in 0.1% Direct Red 80 (Sigma, 365548) in 1.2% picric acid and rinsed in 0.5% acetic acid. Elastic staining was performed following the manufacturer's instructions (elastic stain kit, Sigma HT25A-1KT). Images were acquired with a Pannoramic MIDI Slide scanner (3D HISTECH,

Budapest, Hungary). Polarised images were taken with a Leica DM5500 microscope.

## Whole-mount immunostaining

Embryos were fixed overnight in 4% PFA, washed in 1x PBS followed by dissection of the head. Tissues were permeabilised in 1x PBS, 0.2% Triton X-100 and incubated in proteinase K (5 µg/ml). The heads were then rinsed in glycine and blocked in a solution containing 0.5% goat serum, 0.2% BSA, and 0.2% Triton X-100. Anti-Histone H3 (phospho S10, ab14955) antibodies were diluted by a factor 1500 in blocking solution and samples were incubated overnight at room temperature. Incubation with secondary antibodies (anti-MS-Alexa Fluor 555; 1/500 dilution) was performed during 3 hr at room temperature. Finally, nuclei were stained during 1 hr using Nuclear Green (ab138905) at dilution 1/1500. Z-stacks were acquired using a LSM700 confocal microscope. Nuclei and pH3$^+$ cells were segmented using the Imaris 'spot' tool. Areas containing cells were measured using ImageJ (*Schindelin et al., 2012*) and cell densities were computed. Cell division rates were calculated as the ratio of pH3$^+$ cells over the total number of nuclei.

## 3d reconstructions of embryos

Embryos were fixed in 4% PFA at +4°C overnight, washed in 1X PBS, dehydrated through a methanol series (30%, 50%, 70%, 100%; methanol diluted with 1X PBS) and stored at −20°C. High-resolution episcopic microscopy (HREM; Indigo Scientific, Herts, UK) was performed using the JB4/dye embedding mix (including eosinB and acridine orange) following the manufacturer protocol. Embryos were placed in the mix for overnight polymerisation at room temperature followed by baking for 24 hr at 95°C and sectioning at 3.5 µm. The pixels defining the skin *versus* the underlying connective tissue were separated on HREM 2D images by applying the 2-means clustering algorithm in Python and aligned using the Amira software (ThermoFisher Scientific, Oregon, USA). Then, 3D measurements were performed using Meshlab (*Vcli-, 2011*) and a C ++ based semi-automatic tool (*Milinkovitch et al., 2013*).

## Young's modulus estimation

Fresh embryos were dissected in 1x PBS and fixed and submerged on a home-made support. Nanoindentations were performed using a pre-calibrated FemtoTools FT-S100 Microforce Sensing Probe. Depth-dependant Young's moduli were calculated using unloading curves (*Oliver and Pharr, 2004*) and assuming that the surface of the frill consists of a sheet of skin adherent to the underlying loose connective tissue. Multiple models have been suggested for proper extraction of intrinsic material properties of two layers systems (*Menčík et al., 1997*).

We use a simple closed-form equation (*Jung et al., 2004*) for predicting the elastic moduli of the skin ($E_{skin}$) and of the substrate ($E_{substrate}$) based on the relation between the penetration depth $d$ and the estimated Young's modulus E. The basic assumption of this approach is that the elastic and plastic responses of the system change progressively from skin-dominated to substrate-dominated as the values of $d$ increases (*Figure 11A*). More explicitly, we assume that the Young's modulus E can be written as a simple power law function of the following form:

$$E = E_{substrate}(E_{skin}/E_{substrate})^L$$

where

$$L = 1/(1 + A(d/T)^B)$$

and positive coefficients A and B are calculated using a nonlinear curve-fitting procedure in MATLAB for a range of skin thickness T = 27–72 µm (*Figure 7C*). Given that $E{\rightarrow}E_{skin}$ when $d/T{\rightarrow}0$ and $E{\rightarrow}E_{substrate}$ when $d/T{\rightarrow}\infty$, we estimate $E_{skin}$ = 11'440 ± 200 Pa and $E_{substrate}$ = 220 ± 7 Pa at E26 while $E_{skin}$ = 36'880 ± 20 Pa and $E_{substrate}$ = 228 ± 14 Pa at E45 (*Figure 11B*). This result indicates that the skin modulus is increasing by a factor >3 between E26 and E45 while the substrate's stiffness remains approximately constant.

To test these results, we fix the substrate Young's modulus to the values estimated above with the empirical model of *Jung et al. (2004)*, and use the analytical solution derived by *Gao et al. (1992)* to find the best fit (in terms of mean squared error) of both the skin thickness and skin Young's modulus. These analyses yield values of T = 50 µm and $E_{skin}$ = 17'800 Pa at E26, and of

T = 45 µm and $E_{skin}$ = 38'550 Pa at E45. Hence, the analytical model confirms that the skin thickness remains approximately constant, and that the skin modulus increases substantially (by a factor of about 2.2), between E26 and E45. The somewhat different values of skin stiffness at E26 between empirical and analytical estimates might be due to the low number of data points at small indentation depths.

Note also that there is no consensus in the literature on the best approach for estimating the Young's modulus of biological soft tissues. Indeed, parameters, such as the presence of adhesion forces between the nano-indentation probe and the biological tissue (**Kontomaris, 2018**), but also capillary forces at the air-water interface when measuring submerged samples (**Boots et al., 2019**), can bias the experimental estimates. More fundamentally, the potential anisotropy of biological tissues in terms of their viscoelastic and plastic properties can differentially affect specific methods, for example tensile measurements in multiple kinds of soft biological tissues consistently yield larger Young's modulus estimates than those estimated with indentation methods (**McKee et al., 2011**). Given that all our measurements were performed with the same method in the same conditions, we think that our estimates of the skin-to-substrate stiffness ratio (*i.e.*, the most important parameter for our simulations) are valid.

## Physical analogs

The PDMS elastomer and curing agent (Sylgard 184 Silicone Elastomer Kit) were mixed at a 10:1 vol ratio with the addition of a small amount of green paint (0.03 ml per 100 ml of PDMS). The mixture was placed under vacuum (to avoid bubbles) in a Petri dish for about 30 min before being cured at 50℃ overnight. After cooling, the PDMS sheet was removed from the Petri dish and placed between two wooden planks that were then tightly screwed to each other. The PDMS was swelled by placing the whole system into 98.5% Hexane (Sigma-Aldrich) for 15 min at room temperature.

## Numerical model and computational methods

To study ridges formation under homogeneous growth of the frill constrained at the neck-frill boundary, we constructed a tetrahedral GPU-based numerical model in three dimensions. We assume that the two frill lobes (attached by the central crease) grow independently of one another and that the tissue material is neo-Hookean with volumetric strain energy density

$$W = \frac{\mu}{2}\left[Tr\left(FF^T\right)J^{-2/3} - 3\right] + \frac{K}{2}(J-1)^2$$

where $\mu$ and $K = \alpha.\mu$ are the shear and bulk moduli, respectively, $F$ is the deformation gradient, $J = det(F)$, and $\alpha$ can be defined, using the Poisson's ratio v, by $\alpha = (2+2\nu)/(3-6\nu)$.

We use a custom finite element method, similar to the one described by **Tallinen et al. (2016)** to minimise the elastic energy of the system. Every tetrahedron is defined by a matrix $A = [x_1 \quad x_2 \quad x_3]$, where $x_1$, $x_2$, and $x_3$ are vectors with the origin in the same vertex of the tetrahedron. Following **Jin (2014)** and **Rodriguez et al. (1994)**, we define a growth tensor $G$ of a tetrahedron as $G(\hat{A})$, where $\hat{A}$ describes the tetrahedron in a stress-free initial configuration. The growth tensor is defined by $G = G_sG_c$, where the tensor $G_s = g_sI + (1 - g_s)NN^T$ represents the expansion g(s) perpendicular to the surface normal N in the initial configuration, and the tensor $G_c = I + (g_c - 1)CC^T$ defines the growth of the central crease g(c) in the direction C (**Figure 6D** and **Figure 8A**). Values $g_s$ and $g_c$ are calculated using the generalised logistic function

$$g_i = 1 + \frac{g(i) - 1}{1 + 0.25e^{-100(T-x_i)}}$$

where T is the thickness of the anterior skin sheet of the frill, and $x_i$ is the distance to the surface of the frill for $x_s$ (pink in **Figure 8B**) or to the central crease for $x_c$ (purple in **Figure 8B**). This implies that growth can vary from one element to another which might result in configurations that are not physically attainable (when neighbouring elements do not fit together after growth). To make the volume elements compatible, we introduce the elastic deformation tensor $F(\hat{A})$ that defines how the elements change their shapes to insure continuity inside the tissue. The product of the growth and deformation tensors results in the deformation gradient tensor $A = FG$ that maps the stress-free state $\hat{A}$ before the growth to the stressed state $A = FG\hat{A}$ after growth.

To minimise the elastic energy of the system, at each time step, we obtain the Cauchy stress tensor

$$\sigma = \frac{1}{J}\frac{\partial W}{\partial F}F^T$$

and the surface traction of each deformed face $s_i = -\sigma n_i$ (i = 1,2,3,4), where $n_i$ are normals with lengths proportional to the areas of the deformed faces. Nodal forces are obtained by distributing the traction of each face equally to its three vertices. Finally, the energy of the system is minimised by using damped second-order dynamics

$$\nu(t+\Delta t) = \nu(t) + \frac{f(t) - \gamma\nu(t)}{m}\Delta t$$

$$x(t+\Delta t) = x(t) + \nu(t+\Delta t)\Delta t$$

where the time step $\Delta t = 0.01a/\sqrt{K}$, and $m = a^3$ represents a node mass, where $a$ is the average distance between neighbouring nodes in the initial configuration. The parameter $\gamma$ is the viscous damping factor set to $10m$ in the beginning of the simulation and reduced progressively to speed up convergence. The vectors $f$, $v$ and $x$ are the force, velocity and position of the node, respectively. The simulations are assumed to be at steady state when the elastic energy of the system stabilises (i. e., its value doesn't change by >0.1% during $10^5$ iterations) and the maximum node displacement is $<10^{-6}$.

We also implemented a quasi-static approach using a discrete-time formulation. More explicitly, the elastic energy of the system is minimised at each time step $t_i \in \{1, 2, \ldots, T_{max}\}$, that is for successive values of growth g($t_i$) corresponding to the measured growth values between time steps $t_{i-1}$ and $t_i$. This procedure is performed simultaneously for the surface and the central crease.

To ensure that the frill is a symmetrical structure, the nodes of the central crease can move only along the vector C in one-layer simulations (*Figure 8D*), while they are imposed a zero-displacement boundary condition in the normal direction on the central-crease plane in multi-layer simulations (*Figure 10A*, central panel).

## Acknowledgements

We thank Christoph Bauer, Jérôme Bosset, Déborah Paolucci, António F Martins, Luis Miguel De Oliveira, Carine Langrez and Athanasia Tzika for technical advices. We thank Adrien Debry and Florent Montange for animal care and maintenance.

## Additional information

### Funding

| Funder | Grant reference number | Author |
|---|---|---|
| Swiss National Science Foundation | 31003A_140785 | Michel C Milinkovitch |
| SystemsX | EpiPhysX | Michel C Milinkovitch |
| Swiss National Science Foundation | CR32I3_162743 | Michel C Milinkovitch |
| Swiss National Science Foundation | 31003A_179431 | Michel C Milinkovitch |

The funders had no role in study design, data collection and interpretation, or the decision to submit the work for publication.

### Author contributions

Sophie A Montandon, Data curation, Formal analysis, Validation, Investigation, Methodology, Writing—original draft; Anamarija Fofonjka, Data curation, Software, Formal analysis, Validation,

Investigation, Visualization, Methodology, Writing—original draft; Michel C Milinkovitch, Conceptualization, Resources, Formal analysis, Supervision, Funding acquisition, Validation, Investigation, Methodology, Writing—original draft, Project administration, Writing—review and editing

### Author ORCIDs
Michel C Milinkovitch (iD) https://orcid.org/0000-0002-2553-0724

### Ethics
Animal experimentation: Maintenance of, and experiments on animals were approved by the Geneva Canton ethical regulation authority (authorisations GE/82/14 and GE/73/16) and performed according to Swiss law. These guidelines meet international standards.

### Decision letter and Author response
Decision letter https://doi.org/10.7554/eLife.44455.020
Author response https://doi.org/10.7554/eLife.44455.021

## Additional files

### Supplementary files
• Transparent reporting form
DOI: https://doi.org/10.7554/eLife.44455.018

### Data availability
All data needed to evaluate the conclusions of this study are included in the manuscript and supporting files. Simulation code is available via Bitbucket (https://bitbucket.org/afofonjka/hyperelastic_growth_simulator/src/master/). The instructions of how to use the program are provided in the README file.

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
