## [Decision Letter]

Thank you for submitting your article "Elastic instability during growth of the hyoid branchial ectoderm causes folding of the frilled dragon erectile ruff" for consideration by *eLife*. Your article has been reviewed by David Hu (Reviewer #2) and Pierre Haas (Reviewer #3), and the evaluation has been overseen by a Reviewing Editor and Diethard Tautz as the Senior Editor.

The reviewers have discussed the reviews with one another and the Reviewing Editor has drafted this decision to help you prepare a revised submission.

Summary:

Montandon, Fofonjka, and Milinkovitch present a detailed study of the formation of the ruff of the frilled dragon, combining experiment, physical analog experiments, and simulations of an elastic model. This is a beautiful study that carefully explores several layers of progressively more complex models of the ruff to suggest that the appearance of folds in the ruff is caused by an elastic instability.

The reviewers found the manuscript to be generally well written, although they occasionally found the descriptions slightly confusing: for example, the manuscript sometimes refers to the whole ruff as the "frill", sometimes to each lobe as the "frill", as in the first paragraph of the subsection “The morphology of the frill”, where the frill is said to pleat into three (rather than six) convex ridges. Perhaps using "lobe" more often could clarify these issues.

Essential revisions:

We do think that the title needs to be toned down (to bring the strength of the conclusion more in line with the Discussion), especially given that very recent work has shown that the differential-growth mechanism, although producing brain-like folds, is inconsistent with the folding of the cerebellum (Engstrom et al., Phys Rev. X 8, 041053, 2018). The title is also quite a mouthful, involving many unfamiliar terms in conjunction (e.g. "frilled dragon erectile ruff"). Some simplification would be in order.

The reviewers found the paper at times to be difficult to read. The biological sections could have more terms defined and pointed out in the diagrams. The numerical sections also do not clearly state some of the variables. It would also be helpful if the authors line-numbered this paper for the reviewers. Here isa list of specific issues:

1) Referencing previous work: The authors only cite two references (Savin et al., 2011; Tallinen et al., 2016) on differential growth in morphogenesis. Especially for the brain, folding by differential growth has a long history going back at least 40 years, so previous work should be more carefully referenced. Similarly, in the description of the model, the multiplicative decomposition of the deformation gradient could be put into a clearer context by adding some references (also notice that F is the elastic deformation gradient only, not the full deformation gradient as stated in the text). On a related point, the biophysical significance of their work could be emphasised further if the authors stated more clearly that this curtain instability (geometrically constrained at the boundary) is different from folding of the brain or the gut (geometrically constrained all over the surface).

2) Measurement of elastic moduli: We are not convinced by the measurements of the elastic moduli in Figure 11: (a) the reported values are extrapolations in a range for which there are no measurements; (b) most measurements were taken at an indentation depth for which the moduli have essentially become constant, while no measurements were taken at indentation depths smaller than that corresponding to the inflection point of the curve; (c) the functional form from Jung et al., 2011, used to fit the data is empirical. We think that the authors should state these limitations clearly in the main text, and tone down the strength of their inference that the experimental stiffness ratio is consistent with the theoretical range.

3) Is it possible to make the qualitative comparison of experiment and theory (based on the number of folds that form) more quantitative? For example, do the amplitudes of the folds produced by the model agree with experimental amplitudes? This might help to further constrain the experimental parameters. Using the model in this way to access information not directly accessible by experiments would considerably strengthen the biophysical impact of the analysis. On a related point, the authors estimate growths g(s), g(c), g(o) from geometrical measurements. These observations are however the superposition of actual growth and elastic deformations. It may be worth checking how large the elastic contributions are (by computing the linear surface growth resulting in the model from these imposed growths). Similarly, is there any indication of pre-stressing in the material? While we suppose this could be checked by cutting experiments, we are not asking for these experiments to be carried out, but think it might be worth to point out this possibility.

4) Introduction, second paragraph, the CBI bones, the grey's cartilages, the hyoid apparatus, the hyoid arch, entoglossal process – these should be clearly labelled in a schematic diagram. The dorsal, ventral and lateral views of the embryo are slightly helpful, but a full sketch of the frill and labelling of all its parts would be even better. We struggled with the biology sections, and so will most readers. A diagram would help bridge the physics and biology worlds that necessarily meet in this paper.

5) Subsection “Morphogenesis of the *Chlamydosaurus* neck frill”: E23 = 23 days post-oviposition. This was a useful definition of terms, and it would be great to have more of that throughout the paper.

6) Subsection “Incomplete fusion of BA2 allows the development of a neck skin fold in Episquamata and of the frill in *Chlamydosaurus*”. ‘Bronchial arch’ needs to be defined in a figure, including the four BAs.

7) Subsection “Incomplete fusion of BA2 allows the development of a neck skin fold in Episquamata and of the frill in *Chlamydosaurus*” discusses why other reptile species do not develop frills. However, it was hard to read for non-biologists. Figures or other definitions would be useful here.

8) The authors should avoid using the word integrate, because it is too similar to the math integration.

9) Subsection “Numerical simulations”, last paragraph: the g(s) terminology is confusing. It may have been defined somewhere but it is not easy to find. It should be clearly defined and its use in the aforementioned paragraph should avoid the use of this terminology or re-define it again.

10) Subsection “Numerical simulations”, last paragraph. It is nice that the authors can use the equation of the wrinkle wavelength. A schematic labelling μ_s_

and μ_k_ would be useful for readers who haven't seen this equation before.

11) Subsection “Numerical simulations”, last paragraph. The authors give skin elastic modulus and loose connective tissue modulus. Do these refer to μ_s_ and μ_k_? The authors should say so explicitly. They should also state if the equation they provide indeed gives the correct wrinkle wavelength.

12) Figure 1. Is the yellow hyoid apparatus part of the frill? it would be nice to more clearly relate Figures 1A and 1C.

13) Figure 2 is hard to interpret for non-biologists. What cross section is shown here? It would be good to show a schematic labelling what is cut. This goes for all figures A-M.

14) Figure 3. The dorsal view is not that useful. Are the images from the same embryo?

15) Figure 8 Again g(s) and g(c) is used here but is not labelled in the figure and not easy to understand. Figure 8B does not have a vertical axis.

16) It would be nice to know how the simple wavelength equation varies from numerical results.

17) Figure 8C legend needs to clearly define blue and red lines as well as the units of relative growth.

---

## [Author Response]

[…] The reviewers found the manuscript to be generally well written, although they occasionally found the descriptions slightly confusing: for example, the manuscript sometimes refers to the whole ruff as the "frill", sometimes to each lobe as the "frill", as in the first paragraph of the subsection “The morphology of the frill”, where the frill is said to pleat into three (rather than six) convex ridges. Perhaps using "lobe" more often could clarify these issues.

We modified the text to better indicate when we speak about the whole frill or about one lobe of the frill.

Essential revisions:We do think that the title needs to be toned down (to bring the strength of the conclusion more in line with the Discussion), especially given that very recent work has shown that the differential-growth mechanism, although producing brain-like folds, is inconsistent with the folding of the cerebellum (Engstrom et al., Phys Rev. X 8, 041053, 2018). The title is also quite a mouthful, involving many unfamiliar terms in conjunction (e.g. "frilled dragon erectile ruff"). Some simplification would be in order.

We now propose two alternative titles. The one we strongly prefer is “How the dragon got its frill”, the second is “Elastic instability during branchial ectoderm development causes folding of the *Chlamydosaurus* erectile frill”.

The reviewers found the paper at times to be difficult to read. The biological sections could have more terms defined and pointed out in the diagrams. The numerical sections also do not clearly state some of the variables. It would also be helpful if the authors line-numbered this paper for the reviewers.

Pages are now line-numbered and we provide a specific solution to each specific comment (see below).

Here is a list of specific issues:1) Referencing previous work: The authors only cite two references (Savin et al., 2011; Tallinen et al., 2016) on differential growth in morphogenesis. Especially for the brain, folding by differential growth has a long history going back at least 40 years, so previous work should be more carefully referenced.

We added Nelson, 2016, as a general reference as well as Richman et al., 1975 and Karzbrun, 2018, for the folding of the brain. The reference of Engstrom et al. is additionally introduced in the Discussion (see below).

Similarly, in the description of the model, the multiplicative decomposition of the deformation gradient could be put into a clearer context by adding some references (also notice that F is the elastic deformation gradient only, not the full deformation gradient as stated in the text).

We now better describe and contextualise the model in the Materials and Methods section.

On a related point, the biophysical significance of their work could be emphasised further if the authors stated more clearly that this curtain instability (geometrically constrained at the boundary) is different from folding of the brain or the gut (geometrically constrained all over the surface).

We substantially modified the text (just before the “Numerical simulations” subsection) to emphasise the difference between curtain instability and folding of the brain or gut.

2) Measurement of elastic moduli: We are not convinced by the measurements of the elastic moduli in Figure 11: (a) the reported values are extrapolations in a range for which there are no measurements; (b) most measurements were taken at an indentation depth for which the moduli have essentially become constant, while no measurements were taken at indentation depths smaller than that corresponding to the inflection point of the curve; (c) the functional form from Jung et al., 2011, used to fit the data is empirical. We think that the authors should state these limitations clearly in the main text, and tone down the strength of their inference that the experimental stiffness ratio is consistent with the theoretical range.

To test our results, we fixed the substrate Young’s modulus to the values estimated with the empirical model of (Jung et al., 2011), and we use the analytical solution derived by (Gao et al., 1992) to find the best fit (in terms of mean squared error) of both the skin thickness and skin Young’s modulus. These new analyses (now introduced in the revised manuscript) yield values of T = 50µm and E_skin_ = 17.8 KPa at E26, and of T=45 µm and E_skin_ = 38.55 KPa at E45. Hence, this analytical model confirms that the skin thickness remains approximately constant, and that the skin modulus increases substantially between E26 and E45. In addition to this new analysis (introduced in the third paragraph of the subsection “Young’s modulus estimation”), we also underline now the discrepancy between estimates from the empirical *versus* the analytical model at E26, and we explain that it might be due to the low number of data points at small indentation depths. Finally, in the last paragraph of the aforementioned subsection, we introduce an additional cautionary note regarding the difficulties associated with estimating stiffness of biological tissues, because of probe-tissue adhesion forces, capillary forces at the air-water interface, as well as because of potential anisotropy of the biological material properties.

3) Is it possible to make the qualitative comparison of experiment and theory (based on the number of folds that form) more quantitative? For example, do the amplitudes of the folds produced by the model agree with experimental amplitudes? This might help to further constrain the experimental parameters. Using the model in this way to access information not directly accessible by experiments would considerably strengthen the biophysical impact of the analysis.

We compare now the experimental amplitudes of the ridges with those obtained with the model and discuss the possible causes of discrepancy (subsection “Numerical simulations”, last paragraph).

On a related point, the authors estimate growths g(s), g(c), g(o) from geometrical measurements. These observations are however the superposition of actual growth and elastic deformations. It may be worth checking how large the elastic contributions are (by computing the linear surface growth resulting in the model from these imposed growths).

To measure the elastic contribution in our numerical simulations we measured the linear surface increase at steady state and obtained 1.27 for the value of growth equal to 1.3 (with a stiffness ratio of 100). This information is now introduced in the revised manuscript (subsection “Numerical simulations”, fifth paragraph).

Similarly, is there any indication of pre-stressing in the material? While we suppose this could be checked by cutting experiments, we are not asking for these experiments to be carried out, but think it might be worth to point out this possibility.

We now indicate this possibility in the revised manuscript (subsection “Numerical simulations”, last paragraph).

4) Introduction, second paragraph, the CBI bones, the grey's cartilages, the hyoid apparatus, the hyoid arch, entoglossal process – these should be clearly labelled in a schematic diagram. The dorsal, ventral and lateral views of the embryo are slightly helpful, but a full sketch of the frill and labelling of all its parts would be even better. We struggled with the biology sections, and so will most readers. A diagram would help bridge the physics and biology worlds that necessarily meet in this paper.

We added a diagram of the frill and hyoid apparatus in Figure 1. We modified the main text and figure legend accordingly.

5) Subsection “Morphogenesis of the Chlamydosaurus neck frill”: E23 = 23 days post-oviposition. This was a useful definition of terms, and it would be great to have more of that throughout the paper.

We now indicate that ‘Ex’ means ‘embryonic day x’.

6) Subsection “Incomplete fusion of BA2 allows the development of a neck skin fold in Episquamata and of the frill in Chlamydosaurus”. ‘Bronchial arch’ needs to be defined in a figure, including the four BAs.

We added in Figure 4 a diagram of the branchial arches and their relative positions. We modified the main text and figure legend accordingly.

7) Subsection “Incomplete fusion of BA2 allows the development of a neck skin fold in Episquamata and of the frill in Chlamydosaurus” discusses why other reptile species do not develop frills. However, it was hard to read for non-biologists. Figures or other definitions would be useful here.

All explanations now clearly refer to Figures 4, 5 and Figure 5—figure supplement 1. Hopefully, the diagram of Figure 4 makes these explanations much easier for non-biologists.

8) The authors should avoid using the word integrate, because it is too similar to the math integration.

To avoid ambiguity with the mathematical term, we now replace the word ‘integrate’, throughout the text, with an appropriate alternative (incorporate, apply, etc.).

9) Subsection “Numerical simulations”, last paragraph: the g(s) terminology is confusing. It may have been defined somewhere but it is not easy to find. It should be clearly defined and its use in the aforementioned paragraph should avoid the use of this terminology or re-define it again.

We now much better define the terminology in the subsection “Numerical simulations”.

10) Subsection “Numerical simulations”, last paragraph. It is nice that the authors can use the equation of the wrinkle wavelength. A schematic labelling μ_s_ and μ_k_ would be useful for readers who haven't seen this equation before.

We now much better indicate the correspondence between μ_s_and μ_k_ and the biological tissues.

11) Subsection “Numerical simulations”, last paragraph. The authors give skin elastic modulus and loose connective tissue modulus. Do these refer to μ_s_ and μ_k_? The authors should say so explicitly. They should also state if the equation they provide indeed gives the correct wrinkle wavelength.

As indicated above, we now better define μ_s_and μ_k_in the context of the biological tissues, and we also discuss the differences of wavelengths obtained with the equation *versus* the simulations.

12) Figure 1. Is the yellow hyoid apparatus part of the frill? it would be nice to more clearly relate Figures 1A and 1C.

The new diagram in Figure 1 relates the hyoid apparatus and frill.

13) Figure 2 is hard to interpret for non-biologists. What cross section is shown here? It would be good to show a schematic labelling what is cut. This goes for all figures A -M

Figure 1E indicates now the positions of the sections in Figure 2A-F. The positions of the other sections of Figure 2 are also indicated in Figure 2D and 2F.

14) Figure 3. The dorsal view is not that useful. Are the images from the same embryo?

We have now removed the dorsal views and indicate that the frontal and lateral views are of the same embryo.

15) Figure 8 Again g(s) and g(c) is used here but is not labelled in the figure and not easy to understand. Figure 8B does not have a vertical axis.

g(s) and g(c) are now defined in the subsection “Numerical simulations” (i.e., where Figure 8 is referred to). We also modified Figure 8B legend to define again g(s) and g(c). Figure 8B does not require a vertical axis because we vary only the stiffness ratio (horizontal axis) and look at the resulting geometry (number of folds).

16) It would be nice to know how the simple wavelength equation varies from numerical results.

This is now discussed in the text (subsection “Numerical simulations”).

17) Figure 8C legend needs to clearly define blue and red lines as well as the units of relative growth.

Relative growth does not have units because it is relative. We improved the Figure 8B and 8C legend to make that point clear.